

# Controls on spatial and temporal variability of soil moisture across a heterogeneous boreal forest landscape

Francesco Zignol[1,2], William Lidberg[1], Caroline Greiser[1,2,3], Johannes Larson[1], Raúl Hoffrén[4], and Anneli M. Ågren[1]

[1]Department of Forest Ecology and Management, Swedish University of Agricultural Sciences, Umeå, 90183, Sweden
[2]Bolin Centre for Climate Research, Stockholm University, Stockholm, 10691, Sweden
[3]Department of Physical Geography, Stockholm University, Stockholm, 10691, Sweden
[4]Department of Geography and Land Management, University of Zaragoza, Zaragoza, 50009, Spain

*Correspondence to*: Francesco Zignol (francesco.zignol@slu.se)

**Abstract.** In the light of climate change and biodiversity loss, modeling and mapping soil moisture at high spatiotemporal resolution is increasingly crucial for a wide range of applications in Earth and environmental sciences, particularly in areas like boreal forests where comprehensive soil moisture datasets are scarce. Soil moisture, though a small fraction of Earth's water, plays a fundamental role in terrestrial ecosystem dynamics, influencing meteorological processes, plant health, soil biogeochemistry, groundwater fluctuations, and nutrient exchanges at the land-atmosphere interface. However, understanding and modeling soil moisture dynamics is extremely complex due to the non-linear interplay of numerous physical and biological processes, the large number of drivers involved, and the wide range of spatial and temporal scales at play. Here, we focused on a boreal forest landscape in northern Sweden, where we monitored surface soil moisture with dataloggers at 82 locations during the 2022 vegetation period. We described spatial patterns and temporal fluctuations of soil moisture, we explored the relationships between the observed variations in soil moisture and a vast array of environmental and meteorological factors from multiple sources at varying spatial resolutions and temporal scales, and we tested how these relationships changed over time. Soil properties, topographical features, vegetation characteristics, and land use/land cover were all important contributors of spatial variations in soil moisture, suggesting that current soil moisture maps primarily relying on terrain indices could benefit from integrating this diverse range of information. Moreover, different spatial resolutions and user-defined thresholds of these indices largely affected the performance of the predictions, indicating that topographic proxies for soil moisture should be evaluated for the specific area of interest. Hydrological and meteorological conditions over five to seven days preceding soil moisture measurements were essential in explaining daily soil moisture fluctuations, and influenced the predominant mechanisms governing the spatial distribution of soil moisture. Our findings contribute to advancing physically based land surface and hydrological models, developing machine learning models for predicting spatiotemporal variability in soil moisture, and ultimately generating digital dynamic soil moisture maps for forest management and nature conservation.



## 1 Introduction

Soil moisture, often referred to as the water content within the soil, is a key component in modulating terrestrial ecosystem dynamics, playing a crucial role in the water, energy, and biogeochemical cycles at the interface between the atmosphere and the land surface (Seneviratne et al., 2010; Ochsner et al., 2013). In boreal forests, soil moisture has been proven to affect tree growth (Sikström and Hökkä, 2016; Van Sundert et al., 2018; Larson et al., 2024), influence soil nitrogen availability and, in turn, needle production (Nogovitcyn et al., 2023), and control the distribution of soil organic carbon stocks (Larson et al., 2023). Modeling the soil moisture state, along with its spatial and temporal fluctuations, is essential for numerous Earth and environmental sciences applications, such as weather forecasting, water resource management, forest fire prediction, forest soil trafficability, sustaining ecosystem services, and monitoring ecosystem response to climate change (Babaeian et al., 2019; Peng et al., 2021; Schönauer et al., 2023). Spatial heterogeneity in soil moisture is a key factor in providing diverse habitats, thereby promoting biodiversity (McLaughlin et al., 2017). Temporal variations in soil moisture also influence ecosystem composition, with different species communities depending on more stable or variable soil moisture conditions (Kemppinen et al., 2019). Modeling both components of soil moisture variability assumes even greater significance in the context of climate change and biodiversity loss. In order to accurately model soil moisture, however, it is first necessary to gain a comprehensive understanding of the mechanisms and processes shaping both spatial patterns and temporal dynamics of soil moisture. Despite the considerable research in this field, most studies primarily focused on the spatial variability of soil moisture, often neglecting temporal variations (Kopecký et al., 2021; Ågren et al., 2021; Zhao et al., 2021), restricted analysis to specific spatial resolutions or temporal scales, overlooking their effects on soil moisture predictions (de Oliveira et al., 2021; Tyystjärvi et al., 2022; Schönauer et al., 2023), or analyzed a partial subset of soil moisture drivers, while omitting others (Potopová et al., 2016; Ge et al., 2022; Larson et al., 2022). Thus, there is a clear demand for a thorough examination encompassing all potential controls on soil moisture, with a focus on how varying spatial resolutions and temporal scales might affect the prediction of both its spatial pattern and temporal fluctuations.

Factors influencing soil moisture spatiotemporal variability can be classified into five broad groups: topographical features, soil properties, vegetation characteristics, land use/land cover (LULC), and meteorological forcings (Petropoulos et al., 2013; Rasheed et al., 2022). While spatial variations in soil moisture result from the combined effect of multiple types of drivers, most studies have focused on one or two groups (Gwak and Kim, 2017), with topography being considered the most. Due to the ever-higher spatial resolution of digital elevation models (DEMs), such as those derived from airborne light detection and ranging (LiDAR) measurements, researchers have increasingly relied on terrain indices to explain local influences on soil moisture (Murphy et al., 2011; Lidberg et al., 2020; Ågren et al., 2021; Kopecký et al., 2021). However, only a few studies assessed how the spatial resolution of these indices might affect the prediction of soil moisture (Sørensen and Seibert, 2007; Ågren et al., 2014; Larson et al., 2022). Non-topographical factors usually explain at least half of the spatial variability in soil moisture (Western et al., 1999; Baldwin et al., 2017), and should be taken into account to increase the predictive power of terrain indices (Larson et al., 2022; Kemppinen et al., 2023). Some of these drivers include soil texture



(Krauss et al., 2010), soil depth (Tyystjärvi et al., 2022), organic matter content (Amooh and Bonsu, 2015), hydraulic conductivity (Gwak and Kim, 2017), vegetation density (Gwak and Kim, 2017), vegetation type (Gaur and Mohanty, 2013), snow cover (Potopová et al., 2016), tillage (Jonard et al., 2013), and grazing (Zhao et al., 2011). On the other hand, temporal variations in soil moisture are mostly driven by meteorological variables, such as evapotranspiration and precipitation (McMillan and Srinivasan, 2015; Stark and Fridley, 2023), but the relationship between soil moisture and its controlling factors strongly changes depending on the temporal scale considered (Entin et al., 2000; Parent et al., 2006; Chai et al., 2020). A comprehensive investigation of the role of topography, soil, vegetation, LULC, and meteorological variables at different spatial resolutions and temporal scales in explaining soil moisture variability is essential for gaining new insights into the mechanisms and processes driving soil moisture.

Research has demonstrated that the relative importance of controls on soil moisture spatial distribution can also vary with changing soil wetness conditions over time (Famiglietti et al., 1998; Western et al., 2004; Joshi and Mohanty, 2010; Mei et al., 2018; Gao et al., 2020; Wang et al., 2023). At the catchment level, the wet state is dominated by lateral surface and subsurface flows, which are influenced by nonlocal controls, primarily macrotopography. Conversely, the dry state is characterized by vertical water fluxes, such as infiltration and evapotranspiration, which are influenced by local controls, mainly soil properties and vegetation (Grayson et al., 1997; Western et al., 1999; Rosenbaum et al., 2012). In boreal forests and sub-arctic tundra, the relationship between topography and soil moisture is strong after snowmelt but it weakens towards the end of the snowless season when other processes, such as evaporation and transpiration, primarily control soil moisture patterns (Riihimäki et al., 2021; Kemppinen et al., 2023). Similar results emerged from analyses comparing different seasons (Takagi and Lin, 2012) and years (Gaur and Mohanty, 2013), showing that topography explains more variability in soil moisture spatial patterns during the wetter season/year, while soil characteristics play a more prominent role during the drier season/year. However, recent research indicates that topography may have a different relationship with soil moisture under varying wetness conditions. Regardless of catchment steepness, the relative importance of terrain metrics in explaining soil moisture spatial distribution persisted or even increased as catchments became drier (Liang et al., 2017; Kaiser and McGlynn, 2018; Han et al., 2021) or remained low during the wet season (Dymond et al., 2021). Further research is needed to fully understand how the relationship between soil moisture spatial variability and its controls changes in response to different soil wetness conditions. This information holds practical significance for advancing physically based land surface and hydrological models (e.g., Tyystjärvi et al., 2022), as well as evaluating the effectiveness of more recent modeling techniques, such as machine learning, in predicting and mapping soil moisture spatiotemporal variability (e.g., Schönauer et al., 2023).

In this study, we investigated the climatic and environmental factors that determined spatial patterns and temporal dynamics of surface soil moisture measured using 82 data loggers during the 2022 vegetation period across a heterogeneous boreal forest landscape in northern Sweden. By taking advantage of extensive field measurements, high spatial resolution remote sensing data, and existing maps available for the well-studied Krycklan catchment (Laudon et al., 2013, 2021), we were able to analyze a broad range of potential soil moisture predictors. We tested the hypotheses that the spatial resolution of predictors influences the ability to predict spatial variations in soil moisture, and that meteorological conditions preceding the



logger recordings are key to predict its temporal variations. Additionally, we examined whether the relative importance of predictors in explaining spatial variability in soil moisture changes in response to different wetness conditions throughout the study season. With the ultimate purpose of providing insights into the mechanisms and processes that drive the spatiotemporal

variability in soil moisture, we identified three specific aims: (i) to assess how different variables at varying spatial resolutions affect the prediction of soil moisture spatial variability, (ii) to evaluate the relative contribution of numerous meteorological variables at multiple temporal scales in predicting soil moisture temporal variability, and (iii) to investigate how varying soil wetness conditions over time impact the ability to explain spatial variations in soil moisture.

## 2 Material and methods

### 2.1 Study area

The Krycklan catchment covers an area of about 68 km² in northern Sweden (Fig. 1), with elevations ranging between 127 and 372 m a. s. l. (Larson et al., 2022). Soils, lying on a poorly weathered gneiss bedrock, consist primarily of unsorted glacial till (51%) at higher altitudes and postglacial sorted sediments of sand and silt (30%) at lower altitudes (Laudon et al., 2013). In the northern part of the catchment, peat has built up in areas with low topographic relief, typically forming oligotrophic

minerogenic mires (8.7%; Laudon et al., 2021). The landscape is predominantly forested (87.5%), with Scots pine (*Pinus sylvestris*) and Norway spruce (*Picea abies*) as the main tree species (63% and 26%, respectively), and an understory of bilberry (*Vaccinium myrtillus*) and cowberry (*Vaccinium vitis-idaea*) on moss mats of *Hylocomium splendens* and *Pleurozium schreberi* (Laudon et al., 2013). The remaining coverage includes arable land (2.0%), open land (0.9%), lakes (0.8%), and a small fraction of urban land (0.03%; Lantmäteriet, 2023). The area is characterized by a cold temperate humid climate, with a mean annual

temperature of 2.1°C and a total annual precipitation of 619 mm, of which over 30% falls as snow (Larson et al., 2022).

Since the 1980s, the Krycklan catchment has supported research on ecosystem dynamics and forest management with high-quality, long-term climatic, biogeochemical, hydrological, and environmental measurements, making it a unique field infrastructure in boreal forest landscapes (Laudon et al., 2013). It features 11 gauged streams, around 1000 soil lysimeters, 150 groundwater wells, over 500 permanent forest inventory plots, 3 automatic weather stations (Fig. 1), and a 150 m tall ICOS

(Integrated Carbon Observation System) tower (Fig. 1) for measuring atmospheric gas concentrations and biosphere-atmosphere exchanges of carbon, water, and energy (Laudon et al., 2021). Additionally, high-resolution multi-spectral LiDAR measurements and large-scale experiments have been conducted in the Krycklan catchment over the past decade.





**Figure 1.** Overview of the Krycklan catchment showing the locations of the 82 soil moisture monitoring plots, the three automatic weather
stations, and the ICOS tower, with the ERA5-Land grid superimposed. Orthophoto: Lantmäteriet (2021).



## 2.2 Meteorological and environmental data

The extensive field inventory for the Krycklan catchment, combined with remote sensing and modeled data (e.g., from spatial interpolation and data assimilation), enabled us to evaluate a wide range of meteorological and environmental variables as potential predictors of soil moisture, the response variable in our study. We classified predictors into two groups: "spatial" predictors, which were assumed to be temporally static during the study season but varied spatially, and were used to explain the spatial variability in soil moisture (Table 1); and "temporal" predictors, which varied temporally but not spatially across the study area, and were used to explain the temporal variability in soil moisture (Table 2).

### 2.2.1 Response variable: soil moisture

To measure soil moisture, we selected a subset of 82 plots (Fig. 1) from the forest and soil survey grid described in Larson et al. (2022). These plots cover a wide range of soil moisture conditions, from dry ridges to wet peatlands, effectively capturing the full spectrum of soil moisture levels across the Swedish landscape, as documented by the National Forest Inventory (see Fig. 3 in Ågren et al., 2021). For this reason, about half of the plots clustered in the highly heterogeneous landscape in the central part of the catchment (Fig. 1), 350 m apart from each other or strategically close to permanent measurement stations, including the ICOS tower.

At each site, we measured soil moisture content of the upper 14 cm of soil at 15 min resolution using a TOMST TMS logger (Wild et al., 2019). We installed the loggers in June/July 2022 and we downloaded the data in October 2022, covering 92 days for all sites (from July 5th to October 4th). Because the sensor in the TMS logger relies on the time domain transmission method (Wild et al., 2019), we converted the raw signals into volumetric water content using the universal calibration equation presented in Kopecký et al. (2021). We also evaluated the soil-specific conversion functions proposed by Wild et al. (2019), but we found that some of the resulting volumetric water content values were nonsensical (e.g., <0% and >100%), particularly in mires. Consistent with findings from other studies in similar landscapes (e.g., Kemppinen et al., 2023), we concluded that these conversion functions were unsuitable for the soil types in Krycklan, specifically peat soils. Because the conversion did not alter the relative order among sites, we eventually adopted the universal curve for all plots, which produced a more realistic range of volumetric water content values.

We plotted each individual time series and conducted a thorough visual inspection to identify any anomalies. We checked for sudden drops in soil moisture that quickly reversed, as these often indicate potential loss of contact between sensor and soil. We carefully removed potentially erroneous data to ensure the reliability of our dataset. From the 15 min time series of volumetric water content, we calculated the mean daily time series for each plot, which served as response variables in study aim iii) (Table 3). We then aggregated these data to generate two additional datasets: the seasonal average of mean daily values for each plot and the spatially averaged mean daily time series across all sites, used as response variables in study aims i) and ii), respectively (Table 3). For simplicity, when referring to our analysis, we use the term "soil moisture" in lieu of "volumetric water content at a depth of 0-14 cm".



**2.2.2 Spatial predictors: soil, vegetation, topography, and land use/land cover**

In addition to monitoring soil moisture, we collected a vast array of environmental variables for each of the 82 plots (Fig. 1 and Table 1). Field variables were selected from the Krycklan inventory or during our field campaigns, whereas non-field variables were extracted from existing vector and raster maps, LiDAR-derived topographic indices, and other remote sensing products. In the case of topographic indices and normalized difference vegetation index (NDVI), we extracted plot values from layers at different spatial resolutions (0.5, 1, 2, 4, 8, 16, 32, and 64 m for the topographic indices and 0.4, 2, and 30 m for

NDVI) to assess how varying spatial resolutions explained soil moisture spatial variability. We also tested the effect of different user-defined thresholds, specifically two vertical distances (2 and 4 m) for downslope index and six stream initiation thresholds (1, 2, 4, 8, 16, and 32 ha) for depth to water and elevation above stream (for a thorough description of most topographic indices used in this analysis, refer to Lidberg et al., 2020, and Larson et al., 2022). To facilitate the visualization and interpretation of the results, all predictors were subdivided into four groups, namely soil, vegetation, topography, and land use/land cover

(LULC), and 18 color-coded categories (Table 1). Categories encompass analogous variables from distinct sources (e.g., land cover), diverse measures of a common feature (e.g., tree structure), the same variable at different spatial resolutions and/or user-defined thresholds (e.g., depth to water), or a combination of these cases. Note that classes of qualitative variables were treated as independent predictors in this analysis (e.g., soil type). For a detailed description of all variables, consider the corresponding references in Table 1.

**Table 1.** All potential predictors of soil moisture spatial variability evaluated in this study. The 46 predictors are subdivided into four groups and 18 color-coded categories, listed in alphabetical order within each category based on the abbreviation code ("Abbr." column). The table also displays the data source (field, non-field raster (N-field r), non-field vector (N-field v)), data type (qualitative (Ql) vs. quantitative (Qn)), and references. Most field variables were collected using the same protocols as the Swedish national forest and soil inventories (SLU, 2021). Each class of the qualitative variables is considered as a distinct predictor in the analysis. The number of layers of the topographic and

vegetation indices is reported in parenthesis after the predictor name, and it depends on: [1] the spatial resolutions (0.5, 1, 2, 4, 8, 16, 32, and 64 m for the terrain indices and 0.4, 2, and 30 m for the vegetation index); [2] the stream initiation thresholds (1, 2, 4, 8, 16, and 32 ha for depth to water and elevation above stream); and [3] the vertical distances (2 and 4 m for the downslope index). An asterisk denotes the 23 most relevant predictors that are displayed in Fig. 3. The 23 reaming predictors (no asterisk) are included in the Supplement (Fig. S1).

| Group | Category | Name (number of layers) | Abbr. | Source | Type | Reference |
|---|---|---|---|---|---|---|
| | ■ Organic soil | Organic layer thickness * | olt | Field | Qn | SLU (2021) |
| | ■ Soil depth | SGU soil depth map | sd-sgu | N-field r | Qn | SGU (2024a) |
| | ■ Soil moisture | Soil moisture survey * | sms | Field | Qn | Larson et al. (2022) |
| | | Soil survey – loamy sand * | ss-losa | | | |
| | | Soil survey – peat * | ss-pt | | | |
| | | Soil survey – sand | ss-sa | Field | Ql | SLU (2021) |
| | | Soil survey – sandy loam | ss-salo | | | |
| Soil | | Soil survey – silt loam | ss-silo | | | |
| | ■ Soil type | SGU Quaternary deposit map – clay & silt | st-cs | | | |
| | | SGU Quaternary deposit map – moraine | st-mor | | | |
| | | SGU Quaternary deposit map – postglacial sand | st-psa | N-field v | Ql | SGU (2024b) |
| | | SGU Quaternary deposit map – postglacial gravel | st-pgr | | | |
| | | SGU Quaternary deposit map – peat * | st-pt | | | |





| Group | Category | Name (number of layers) | Abbr. | Source | Type | Reference |
|---|---|---|---|---|---|---|
| Topography | ▪ Aspect | Aspect (8) | asp-[1] | | | |
| | ▪ Depth to water | Depth to water (48) * | dtw[2]-[1] | | | |
| | ▪ Downslope index | Downslope index (16) * | di[3]-[1] | | | |
| | ▪ Elevation above stream | Elevation above stream (48) * | eas[2]-[1] | | | |
| | ▪ Landscape wetness Index | Landscape wetness index (8) * | wilt-[1] | N-field r | Qn | Lidberg et al. (2020) |
| | ▪ Plan curvature | Plan curvature (8) * | plc-[1] | | | |
| | ▪ Relative topographic position | Relative topographic position (8) * | rtp-[1] | | | |
| | ▪ Slope | Slope (8) | sl-[1] | | | |
| | ▪ Topographic wetness index | Topographic wetness index (8) * | twi-[1] | | | |
| | ▪ Topography-based map | Soil moisture index map * | smi | N-field r | Qn | Naturvårdsverket (2022) |
| | | SLU soil moisture map * | sm-slu | | | Ågren et al. (2021) |
| Vegetation | ▪ Forest productivity | Site quality index * | sqi | Field | Qn | SLU (2021) |
| | | Biomass above ground | bio | | | |
| | | SLU forest biomass map | bio-slu | N-field r | Qn | Nilsson et al. (2017) |
| | | Normalized difference vegetation index (3) * | ndvi-[1] | N-field r | Qn | Lantmäteriet (2021), Vermote et al. (2016) |
| | | Stem density * | stm | Field | Qn | SLU (2021) |
| | ▪ Species composition | Volume of birch species | bir | Field | | SLU (2021) |
| | | SLU birch map | bir-slu | N-field r | | Nilsson et al. (2017) |
| | | Volume of pine species * | pi | Field | Qn | SLU (2021) |
| | | SLU pine map * | pi-slu | N-field r | | Nilsson et al. (2017) |
| | | Volume of spruce species | spr | Field | | SLU (2021) |
| | | SLU spruce map | spr-slu | N-field r | | Nilsson et al. (2017) |
| | ▪ Tree structure | Canopy openness | co | | | Hederová (2023) |
| | | Basal area weighted mean diameter | dgv | Field | | SLU (2021) |
| | | Volume of all tree species | for | | Qn | |
| | | SLU forest map | for-slu | N-field r | | Nilsson et al. (2017) |
| | | Basal area weighted mean height | hgv | Field | | SLU (2021) |
| | | SLU basal area weighted mean height map | hgv-slu | N-field r | | Nilsson et al. (2017) |
| LULC | ▪ Land cover | Lantmäteriet land map – forest * | lm-for | N-field v | Ql | Lantmäteriet (2023) |
| | | Lantmäteriet land map – mire * | lm-mir | | | |
| | | Land survey – clearcut | ls-cut | | | |
| | | Land survey – forest * | ls-for | Field | Ql | SLU (2021) |
| | | Land survey – mire * | ls-mir | | | |

### 2.2.3 Temporal predictors: meteorological forcings

For the temporal analysis, we selected meteorological variables (Table 2) from three datasets, including reanalysis data from

the land component of the European Centre for Medium-Range Weather Forecasts (ECMWF) Atmospheric Reanalysis Fifth

Generation (ERA5-Land; Muñoz-Sabater et al., 2021), atmospheric data from the ICOS tower (Peichl et al., 2024), and three



automatic weather stations (Svartberget Research Station, 2023a, b, c). For each variable, we generated a single daily time series from July 5th to October 4th, 2022, for the entire catchment by calculating the spatial average between either the three

weather stations or the six ERA5-Land cells covering the Krycklan area (Fig. 1). To evaluate how varying temporal scales explained temporal variability of soil moisture, we created seven additional time series for each variable based on different temporal scales, including the preceding day and the average between 3, 5, 7, 10, 14, and 21 preceding days. All predictors were subdivided into 12 color-coded categories to facilitate the visualization and interpretation of the results. These categories group together analogous variables from distinct sources (e.g., precipitation), any variable measured at different depths (e.g.,

soil water) or heights (air temperature), diverse aspects of the same process (e.g., evaporation), or a combination of these cases.

**Table 2.** All potential predictors of soil moisture temporal variability assessed in this study. The 60 predictors are subdivided into 12 color-coded categories, listed in alphabetical order within each category based on the abbreviation code ("Abbr." column). The table also indicates the unit of measurement, the dataset (ERA5-Land, ICOS tower, or weather stations), and data source (field vs. non-field (N-field)). Whenever

possible, either the sensor height (field data) or the height of the estimated values (ERA5-Land) is reported in parenthesis after the predictor name. An asterisk denotes the 25 most relevant predictors that are displayed in Fig. 4. The 35 reaming predictors (no asterisk) are included in the Supplement (Fig. S2).

| Category | Name (height of sensor or estimated values) | Abbr. | Unit | Dataset | Source |
|---|---|---|---|---|---|
| ■ Air temperature | 2m dewpoint temperature (2 m) | d2m | | | |
| | Skin temperature (0 m) | skt | | ERA5-Land | N-field |
| | 2m temperature (2 m) | t2m | | | |
| | Air temperature (1.7 m) | ta | °C | Weather stations | |
| | Air temperature level 1 (42 m) | ta1 | | | |
| | Air temperature level 2 (30 m) | ta2 | | ICOS tower | Field |
| | Air temperature level 3 (20 m) | ta3 | | | |
| | Air temperature level 4 (10 m) | ta4 | | | |
| ■ Air water | Air relative humidity (32.5 m) * | rh | % | ICOS tower | Field |
| | Skin reservoir content * | src | mm | ERA5-Land | N-field |
| ■ Evaporation | Total evaporation | e | | | |
| | Evaporation from bare soil | ebs | | | |
| | Potential evaporation * | ep | mm, accumulated | ERA5-Land | N-field |
| | Evaporation from the top of canopy * | etc | | | |
| | Evaporation from vegetation transpiration * | evt | | | |
| ■ Heat | Soil heat flux level 1 (0 cm) | sh1 | W/m², accumulated | ICOS tower | Field |
| | Soil heat flux level 2 (5 cm) | sh2 | | | |
| | Surface sensible heat flux (0 m) * | shf | J/m², accumulated | ERA5-Land | N-field |
| ■ Precipitation | Total precipitation * | p | mm, accumulated | ERA5-Land | N-field |
| | Total precipitation (1.5 m) * | pr | | Weather stations | Field |
| ■ Pressure | Air pressure (1.7 m) * | pa | | Weather stations | Field |
| | Surface pressure (0 m) * | sp | hPa | ERA5-Land | N-field |
| | Vapor pressure (1.7 m) | vp | | Weather stations | Field |
| | Vapor pressure deficit (32.5 m) * | vpd | | ICOS tower | |



| Category | Name (height of sensor or estimated values) | Abbr. | Unit | Dataset | Source |
|---|---|---|---|---|---|
| **Radiation** | Forecast albedo | fal | dimensionless, 0-1 | ERA5-Land | N-field |
| | Long wave incoming radiation (50 m) * | lwi | W/m², accumulated | ICOS tower | Field |
| | Long wave outgoing radiation (50 m) | lwo | | | |
| | Surface net solar radiation (0 m) | nsr | J/m², accumulated | ERA5-Land | N-field |
| | Surface net thermal radiation (0 m) * | ntr | | | |
| | Short wave incoming radiation (50 m) * | swi | W/m², accumulated | ICOS tower | Field |
| | Short wave outgoing radiation (50 m) * | swo | | | |
| | Surface thermal radiation downwards (0 m) * | trd | J/m², accumulated | ERA5-Land | N-field |
| **Runoff** | Runoff | ro | mm, accumulated | ERA5-Land | N-field |
| | Surface runoff (0 m) * | sr | | | |
| | Sub-surface runoff | ssr | | | |
| **Soil temperature** | Soil temperature level 1 (0-7 cm below surface) | st1 | °C | ERA5-Land | N-field |
| | Soil temperature level 2 (7-28 cm below surface) | st2 | | | |
| | Soil temperature level 3 (28-100 cm below surface) * | st3 | | | |
| | Soil temperature level 4 (100-289 cm below surface) | st4 | | | |
| | Soil temperature level 1 (-10 cm) | ts1 | | Weather stations | Field |
| | Soil temperature level 2 (-20 cm) | ts2 | | | |
| | Soil temperature level 3 (-30 cm) * | ts3 | | ICOS tower | |
| | Soil temperature level 4 (-50 cm) | ts4 | | | |
| **Soil water** | Soil water content level 1 (-2.5 cm) * | sm1 | % | ICOS tower | Field |
| | Soil water content level 2 (-5 cm) | sm2 | | | |
| | Soil water content level 3 (-10 cm) | sm3 | | | |
| | Soil water content level 4 (-30 cm) | sm4 | | | |
| | Volumetric soil water level 1 (0-7 cm below surface) * | sw1 | | ERA5-Land | N-field |
| | Volumetric soil water level 2 (7-28 cm below surface) | sw2 | | | |
| | Volumetric soil water level 3 (28-100 cm below surface) | sw3 | | | |
| | Volumetric soil water level 4 (100-289 cm below surface) | sw4 | | | |
| **Vegetation** | Leaf area index, high vegetation | lai | m²/m² | ERA5-Land | N-field |
| | Photosynthetic photon flux density below canopy incoming (1.15 m) | pbc | μmolPhotons/m²/s | ICOS tower | Field |
| | Photosynthetic photon flux density diffuse (50 m) | pd | | | |
| | Photosynthetic photon flux density incoming (50 m) * | pi | | | |
| | Photosynthetic photon flux density outgoing (50 m) * | po | | | |
| **Wind** | 10m u-component of wind * | u10 | m/s | ERA5-Land | N-field |
| | 10m v-component of wind | v10 | | | |
| | Wind direction respect to geographic north (34.5 m) | wd | degrees N | ICOS tower | Field |
| | Wind speed (34.5 m) * | ws | m/s | | |

## 2.3 Statistical model

To identify significant predictors of soil moisture, we used orthogonal projections to latent structures (OPLS) analysis, an
enhanced version of partial least-squares regression (PLS; Eriksson et al., 2013). OPLS separates the systematic variation in





the predictors (X) into two parts: a predictive component (horizontal axis) that is directly associated with the response variable of interest (Y) and an orthogonal component (vertical axis) that represents the variation unrelated to Y. This method improves interpretability over ordinary PLS as it allows for identifying key variables for predicting Y while isolating less important variables that contain noise. In this two-dimensional space, positive or negative loadings on the predictive axis denote variables

that are positively or negatively correlated with Y, with stronger correlations as distance from the origin increases. Conversely, loadings on the orthogonal axis, farther from the origin, indicate less correlated variables (i.e., higher noise). In our study, we used soil moisture measurements from dataloggers as the response variable (Y).

We created two types of OPLS models (Table 3). The first type, termed "spatial" OPLS, assessed the role of environmental predictors (soil, topography, vegetation, and LULC) (Table 1) in explaining the observed spatial distribution in

soil moisture through direct plot-by-plot comparison. In these models, all environmental predictors varied across Krycklan but were assumed constant over time. Similarly, the response variable (i.e., soil moisture) was spatially heterogeneous, but only one time step was included in each model (either the seasonal average or the value of a certain day). The second type, termed "temporal" OPLS, evaluated the influence of meteorological predictors (Table 2) on the observed daily variations in soil moisture through direct day-by-day comparison. In this model, all meteorological predictors and the response variable changed

daily but were considered uniform across the study area (i.e., we calculated the spatial average).

For all 94 OPLS models (Table 3), we also calculated the variable importance on projection for the predictive component ($VIP_{predictive}$). These values are normalized such that if each X variable contributed equally to the model, their $VIP_{predictive}$ would be 1. Variables with a $VIP_{predictive}$ value greater than 1 are considered relevant predictors, with higher scores indicating greater predictive power (Eriksson et al., 2013). We processed all the data in R version 4.3.0 (R Core Team, 2023),

we generated all OPLS models and calculated the related $VIP_{predictive}$ scores in SIMCA 17.0, and we drew all the figures using the R ggplot2 package (Wickham, 2016), ArcGIS Pro (Esri Inc., 2023), and Adobe Illustrator (Adobe Inc., 2024).

**Table 3.** All OPLS models developed in this study.

| Aim | Model type | Predictors (X) | Predictors' characteristics | Response variable (Y) | # of models | Figure |
|---|---|---|---|---|---|---|
| i) | Spatial OPLS | Soil Topography Vegetation LULC | Different spatial resolutions and user-defined thresholds | Seasonal average of mean daily values for each plot | 1 | 3 |
| ii) | Temporal OPLS | Meteorological forcings | Different temporal scales | Spatially averaged mean daily time series across all sites | 1 | 4 |
| iii) | Spatial OPLS | Soil Topography Vegetation LULC | Different spatial resolutions and user-defined thresholds | Mean daily value of any day within the season for each plot | 92 | 5 |



## 3 Results

### 3.1 Observed spatial and temporal variability in soil moisture

Analysis of the logger data revealed large spatial variability in both seasonal averages and seasonal standard deviations of soil moisture, ranging from 14% to 56% and 0.4% to 5.6%, respectively (Fig. 2a). Among the 82 sites studied, 15 exhibited an increasing trend in soil moisture over the season, seven a decreasing trend, and the remaining 60 no trend, based on the non-parametric Mann-Kendall test (Mann, 1945; Kendall, 1975) at 95% confidence level (Fig. 2bc). The magnitude of soil moisture change over the entire study period, indicated by the trend Theil-Sen's slope (Sen, 1968), varied between -8.4% and 10% (Fig.

2b, Table S1), whereas the strength of the monotonic association between soil moisture and time, as measured by Kendall's correlation coefficient ($\tau$), ranged from -0.58 to +0.57 (Table S1). Daily peaks in soil moisture were typically associated with major precipitation events, although the magnitude of these peaks and subsequent declines during dry periods varied considerably across locations (Fig. 2c). Conversely, the daily spatial variability (i.e., standard deviation) in soil moisture (black line) exhibited a sharp decline during precipitation occurrences, followed by a steady increase leading up to peaks at the

culmination of subsequent dry periods (bottom part of Fig. 2c). The soil moisture time series from ERA5-Land (brown lines) closely tracked the temporal variability of the sites mean (red line), but underestimated daily soil moisture amounts averaged across all sites (Fig. 2c). Overall, Fig. 2 showed that the 82 sites responded differently to similar weather conditions, and that the spatial variability in soil moisture among all sites is much larger than the temporal variability in soil moisture observed throughout the study season.






**Figure 2.** Spatial and temporal variation of daily mean soil moisture (i.e., volumetric water content) measured by 82 loggers across the Krycklan catchment from July 5th to October 4th, 2022. (a) Displays the seasonal average and standard deviation of the measurements. (b) Shows seasonal trends identified using the Mann-Kendall test at a 95% confidence interval. (c) Presents the time series plot, with logger data grouped by color according to trend type. The graphic includes additional data for comparison: estimates from six ERA5-Land cells covering the catchment (brown lines), spatial mean (red line) and standard deviation (black line) among sites, and mean precipitation across Krycklan derived from weather stations (bottom bar plot). For clarity, refer to Fig. 1 for the locations of the ERA5-Land cells and weather stations. Orthophoto in panels (a) and (b): Lantmäteriet (2021).

## 3.2 Controls on soil moisture variability

OPLS plots served as a means to visualize in two dimensions the relative importance of factors controlling soil moisture variability, with loadings located closer to the horizontal axis (i.e., lower noise) and farther from the vertical axis (i.e., higher predictive power) indicating the most relevant predictors. Variables on the right side of the plot are positively correlated to soil moisture, while those on the left side are negatively correlated. Remote sensing and modeled estimates are represented by circles (raster datasets) or rhombuses (vector datasets), whereas field measurements are displayed as triangles. The size of the symbols is proportional to either the spatial resolution or the temporal scale of the potential soil moisture predictors. Variables are grouped together into color-coded categories to facilitate the reading of the OPLS plots. When multiple spatial resolutions or temporal scales were investigated for a certain variable, its loadings were connected through guides transitioning from high to low resolution or scale, and only the optimal resolution or scale was labelled. The upcoming two sections will focus on outlining the key features of the spatial OPLS plot (Figs. 3 and S1) and the temporal OPLS plot (Figs. 4 and S2), respectively. Due to the large amount of variables analyzed in this study, Figs. 3 and 4 only present the most relevant predictors (VIP$_{predictive}$ greater than 1, marked by an asterisk in Tables 1 and 2), whereas all remaining variables are included in the Supplement (Figs. S1 and S2).

### 3.2.1 Spatial variation

Relative topographic position emerged as the strongest predictor of soil moisture at a 16 m resolution (rtp-16), but its predictive performance decreased at lower and higher resolutions (Fig. 3). Similar to relative topographic position, depth to water and elevation above stream were negatively correlated with soil moisture, with loadings clustered in the bottom-left quadrant (Figs. 3 and S1). These two indices showed reduced performance and increased noise for higher stream initiation thresholds (Fig. S1). However, while coarse resolution (64 m) was optimal for elevation above stream, high resolution (0.5 or 1 m) was preferable for depth to water (Fig S1), with eas1-64 and dtw1-05 overall performing best (Fig. 3). In the top-right quadrant (i.e., positively correlated), topographic wetness index and landscape wetness index were good predictors of soil moisture at their optimal resolutions of 32 m (twi-32) and 4 m (wilt-4), respectively (Fig. 3). At these resolutions, they performed comparably to the soil moisture index map (smi) and the SLU soil moisture map (sm-slu), with the last one exhibiting slightly higher performance (Fig. 3). Plan curvature and downslope index at their optimal vertical distance and/or spatial resolution (di2-32 and plc-32), also positively correlated with soil moisture, showed slightly lower predictive power but introduced less




noise (loadings closer to the origin) (Fig. 3). Other DEM-derived variables, including slope and aspect, were less relevant
predictors (Fig. S1).

As for soil, three field variables – peat soil class (ss-pt), soil moisture classes (sms), and organic layer thickness (olt)
– were robust predictors, showing a positive correlation with soil moisture and low noise (Fig. 3). The peat class from the SGU
soil type map (st-pt) was also positively correlated, yet it explained less variability than the analogous field predictor (i.e., ss-
pt). Both mire (positively correlated) and forest (negatively correlated) LULC classes similarly revealed that the data collected
in the field (ls-mir and ls-for, respectively) provided slightly better results than using information from an existing map (lm-
mir and lm-for, respectively). Finally, the loamy sand class from the soil survey (ss-losa) was, to a lesser extent, an important
predictor, negatively correlated with soil moisture. The remaining soil and LULC variables, whether derived from field
observations or existing maps, performed poorly in predicting soil moisture (Fig. S1).

Among the vegetation-related variables, volume of pine (pi) showed the highest predictive performance, followed by
the normalized difference vegetation index at 2 m resolution (ndvi-2), the site quality index (sqi), the SLU pine map (pi-slu),
and stem density (stm), with pi, sqi, and pi-slu being negatively correlated with soil moisture whereas ndvi-2 and stm being
positively correlated (Fig. 3). While ndvi-2 and pi slightly outperformed, in terms of predictive power, analogous predictors
at courser spatial resolutions (ndvi-30 and pi-slu, respectively), they also introduced more noise. The remaining vegetation
variables exhibited low predictive performance or high noise, therefore resulting less suitable as soil moisture predictors (Fig.
S1).







**Figure 3.** OPLS loading plot showing the relationship between a large array of "spatial" predictors, which vary spatially but remain constant over time, and the mean seasonal soil moisture (July 5 – October 4, 2022). Both the spatial predictors (X-variables) and the determinant (Y-variable) were gathered for 82 sites across the Krycklan catchment (Fig. 1 for the site locations). The spatial predictors, overall describing soil, topography, vegetation, and land use/land cover at each site (grey dotted boxes in the figure legend) were either directly measured in situ (symbolized by triangles) or estimated through remote sensing or modeling techniques (depicted as circles or rhombuses depending on the dataset format). These predictors were organized into 18 color-coded categories (see Table 1; here only 14) to enhance plot readability. Gridded (i.e., raster) predictors are characterized by a certain spatial resolution, represented by the size of the circles. To visualize the effects of spatial resolution, guides connect loadings of the same variable moving from high to low resolutions, with the variable name visible only in correspondence of the optimal resolution (refer to Table 1 for variable labels). High positive and negative loadings on the predictive axis (pq[1]) represent variables that are positively and negatively correlated with the response variable (Y), with stronger correlation further way





from the origin. The orthogonal axis (poso[1]) indicates how much of the variation for each variable was not correlated with the response variable (Y). This figure only shows the 23 most relevant predictors (VIP$_{predictive}$ greater than 1, marked by an asterisk in Table1). If multiple user-defined thresholds were tested for a certain topographic index (i.e., depth to water, downslope index, and elevation above stream), the plot displays only the best-performing one. All remaining variables are included in the Supplement (Fig. S1).

### 3.2.2 Temporal variation

Soil moisture estimates from ERA5-Land and ICOS measurements were understandably the two best predictors of the spatially averaged time series of soil moisture recorded at the 82 study plots (Fig. 4). Their predictive performance was highest when selecting the top soil layer and matching the temporal scale with the response variable (sw1-0 and sm1-0). Most loadings of these two predictors were positively correlated with the determinant (Y), though the strength of the correlation generally decreased and noise increased with longer temporal scales and deeper soil layers (Fig. S2).

The temporal OPLS analysis revealed that the optimal temporal scale for most predictors ranged between 5 and 7 days preceding the datalogger recordings, with predictive performance decreasing for both shorter and longer temporal scales (Fig. 4). Skin reservoir content, which accounts for the water in the vegetation canopy and in a thin layer on top of the soil, at the 7-day scale (src-7), emerged as a strong predictor, positively correlated with soil moisture and associated with minimal noise. Surface air pressure at the 7-day scale (sp-7 and pa-7) was also a robust predictor, showing an inverse correlation with soil moisture. Evaporation from the top of canopy at the 5-day scale (etc-5) lay in the vicinity, yet towards higher noise and lower predictive values.

The remaining variables explaining the temporal variability in soil moisture clustered into three distinct areas (Fig. 4). In the right side of the OPLS plot, therefore indicating a positive relationship with soil moisture, two clusters stood out: air relative humidity (rh-7), surface net thermal radiation (ntr-7), surface sensible heat flux (shf-3), evaporation from vegetation transpiration (evt-7), and potential evaporation (ep-5) in the top quadrant; precipitation (pr-7 and p-5), surface runoff (sr-5), long-wave (i.e., thermal) incoming radiation (lwi-5 and trd-5), and wind speed (ws-5) in the bottom quadrant. The third cluster, located in the bottom-left quadrant, consisted of predictors negatively correlated with soil moisture, including incoming and outgoing short wave radiation (swo-5 and swi-5), incoming and outgoing photosynthetic photon flux density (po-5 and pi-5), vapor pressure deficit (vpd-7), 10m u-component of wind (u10-5), and soil temperature (ts3-21 and st3-21). All air temperature variables, along with other less relevant predictors of soil moisture, are showed in the Supplement (Fig. S2).





**Figure 4.** OPLS loading plot illustrating the relationship between a large array of "temporal" predictors, which do not vary spatially but change over time, and daily mean soil moisture (i.e., volumetric water content) averaged across 82 sites within the Krycklan catchment (refer to Fig. 1 for the site locations). Both the temporal predictors (X-variables) and the determinant (Y-variable) were aggregated at the daily temporal scale from July 5th to October 4th, 2022. The temporal predictors were either directly measured at the ICOS tower or at weather stations within Krycklan (symbolized by triangles) or extracted from the ERA5-Land dataset (depicted as circles). These predictors were organized into 12 color-coded categories (see Table 2; here only 11) to enhance plot readability. All predictors are characterized by a certain temporal scale, represented by the size of the triangles or circles. To visualize the effects of temporal scale, guides connect loadings of the same variable moving from high to low scales, with the variable name visible only in correspondence of the optimal scale (refer to Table 2 for variable labels). High positive and negative loadings on the predictive axis (pq[1]) represent variables that are positively and negatively correlated with the response variable (Y), with stronger correlation further way from the origin. The orthogonal axis (poso[1]) indicates how much of the variation for each variable was not correlated with the response variable (Y). This figure only shows the 25 most relevant predictors (VIP$_{predictive}$ greater than 1, marked by an asterisk in Table2), but the 35 remaining predictors are included in the Supplement (Fig. S2).




### 3.3 Spatial soil moisture variability under different wetness conditions

The relative importance of predictors in influencing spatial soil moisture variability remained relatively consistent over the 2022 vegetation period in the Krycklan catchment, with their VIP$_{predictive}$ values showing little variation throughout the season

(Figs. 5 and S3). The SLU soil moisture map (sm-slu) exhibited the smallest variation among all predictors (seasonal standard deviation of VIP$_{predictive}$: 0.03) (Fig. 5). In contrast, three vegetation-related variables (pi-slu, pi, and ndvi-2) showed the largest variation (seasonal standard deviation of VIP$_{predictive}$: 0.1), reflecting generally better performances in the first half of the season (especially at the turn of July and August) compared to the second half (Figs. 5 and S3).

Most predictors experienced abrupt drops in VIP$_{predictive}$ during intense and/or multi-day precipitation occurrences

(e.g., September 16$^{th}$) (Fig. 5), when the soil moisture variability across all 82 sites was also at its lowest (bottom graphic in Fig. 2c). However, some topographic indices (dtw1-05, eas1-64, and, to a lesser degree, plc-32 and rpt-16) showed increasing predictive power after the beginning of a precipitation event (e.g., July 15$^{th}$ or September 15$^{th}$) (Fig. 5). During drying periods (e.g., between end of August and almost mid-September), the VIP$_{predictive}$ values of the majority of predictors tended to steadily and slowly decrease, except for two notable exceptions: the loamy sand soil class (ss-losa) and the downslope index (di2-32).

**Figure 5.** VIP$_{predictive}$ values of 92 spatial OPLS models generated using mean daily soil moisture over the study season (July 5$^{th}$ – October 4$^{th}$, 2022) as the response variable (Y). The lower section of the figure displays the mean precipitation across Krycklan derived from weather






stations (refer to Fig. 1 for their locations). The spatial predictors, overall describing soil, topography, vegetation, and land use/land cover at each site (grey dotted boxes in the figure legend), were organized into 18 color-coded categories (see Table 1; here only 13) to enhance plot
readability. Color-coded labels on the right side of the figure are ordered according to their VIP$_{predictive}$ on the last day of the study season (October 4, 2022). To avoid clutter and highlight the key findings, only a subset of predictors is presented, but a graphic with all 23 relevant predictors (VIP$_{predictive}$ greater than 1) identified in Fig. 3 is included in the Supplement (Fig. S3).

## 4 Discussion

In this study, we investigated a vast array of climatic and environmental factors controlling spatial patterns and temporal
dynamics of surface soil moisture in a forest boreal landscape in northern Sweden with the purpose of providing new insights into modeling and mapping soil moisture. Specifically, we evaluated the ability of numerous variables extracted from multiple sources, including field measurements, remote sensing retrievals, and modeled data at different spatial resolutions and temporal scales, to predict soil moisture variations recorded during the 2022 vegetation period by 82 dataloggers distributed across the Krycklan catchment. In the sections that follow, we discuss the primary findings from our analysis.

### 4.1 Spatial variation

We found that all four groups of spatial predictors considered in this analysis, namely topographical features, soil properties, vegetation characteristics, and land use/land cover (LULC), played a significant role in explaining spatial variations in soil moisture (Fig. 3). With the advent of LiDAR-derived DEMs at very high spatial resolution, researchers have increasingly used terrain indices, or a combination of them, as a proxy for soil moisture (Kopecký et al., 2021; e.g., Riihimäki et al., 2021;
Winzeler et al., 2022), including the 10-m resolution soil moisture index map (smi; Naturvårdsverket, 2021) and the 2-m resolution SLU soil moisture map (sm-slu; Ågren et al., 2021) that we evaluated in our study. While these maps correlated well with soil moisture measured in the field, our analysis revealed that soil predictors, such as organic layer thickness and soil texture, vegetation-related variables, and land cover information distinguishing between mire and forest were also important. The relevance of integrating soil and terrain information to characterize soil moisture patterns in the context of
hydrological modeling was highlighted by similar studies at the catchment scale (e.g., Baldwin et al., 2017). Previous research demonstrated that soil properties were determinant in controlling soil moisture spatial variance at the hillslope (Wang et al., 2023) and regional (Wu et al., 2020) scales as well. Consistent with other studies (e.g., Sørensen and Seibert, 2007; Ågren et al., 2014; Lidberg et al., 2020; Larson et al., 2022), our analysis also indicated that the performance of any terrain index varied greatly depending on the threshold and resolution considered, with 1-ha stream initiation threshold providing the best results
and 0.5-m spatial resolution being the optimal choice only in one case (i.e., depth to water index). Interestingly, relative topographic position at 16-m resolution (rtp-16) emerged as the best predictor of soil moisture spatial variability, capable of identifying wetter depressions and drier ridges in the landscape (Weiss, 2001). While several examples in the literature demonstrate the importance of this index in soil moisture estimation (e.g., Engstrom et al., 2005; Zhao et al., 2021), it is somewhat surprising that our findings disagree with those of Kemppinen et al. (2018), who found that the topographic wetness
index outperformed relative topographic position in predicting soil moisture in a boreal landscape in northwestern Finland



comparable with our study landscape. Larson et al. (2022), who used five soil moisture classes estimated in the field as response variable (sms predictor in our study; see Table 1 and Fig. 3), observed that relative topographic position was not among the best performing variables in the Krycklan catchment. Therefore, in the pursuit of estimating spatial variability in soil moisture, we advise caution when selecting terrain indices and their spatial resolutions and thresholds, we argue that an enhanced spatial
resolution in topographical data does not necessarily compensate for the absence of soil, vegetation, and LULC information, and we reiterate the importance of soil moisture field measurements to validate predictive models.

## 4.2 Temporal variation

Our research demonstrated that daily soil moisture fluctuations within the Krycklan catchment are strongly influenced by the hydrological and meteorological conditions over five to seven days preceding soil moisture measurements, regardless
of whether these conditions were estimated (ERA5-Land dataset) or measured directly in the field (weather stations and ICOS tower) (Fig. 4). Among other variables, increased soil moisture was correlated with lower air pressure, shortwave radiation, vapor pressure deficit, and evaporation from the top of canopy; conversely, it was associated with higher thermal (longwave) radiation, precipitation, air humidity, evapotranspiration, and wind speed. Averaged conditions over five to seven days for all these variables exhibited the strongest correlation with daily variations in soil moisture in Krycklan, indicating both lagged
and cumulative effects of these processes on soil moisture. Previous research has also highlighted the importance of considering multi-day accumulations and time lags between meteorological drivers and soil moisture response (Williams et al., 2009; Pan, 2012; Li et al., 2024), with most studies focusing on precipitation-soil moisture relationship. Parent et al. (2006) showed that the transfer of energy from precipitation to soil moisture via infiltration, percolation, and redistribution processes mostly occurs over temporal scales ranging between 2 and 14 days. Piao et al. (2009) proved that precipitation frequency can
be a more crucial factor than precipitation amount in shaping soil moisture variations, making it essential to account for the cumulative effect of precipitation over multi-day temporal scales (Ge et al., 2022). Our study identified soil temperature (28-100 cm below surface) as the most notable exception to the optimal temporal scale of five to seven days observed for almost all other relevant predictors. While we found a negative correlation between soil temperature and soil moisture as expected (Aalto et al., 2013), the strongest effects emerged at the 3-week scale, the longest temporal scale considered in our analysis.
Soil temperature, along with air temperature – which performed poorly in our study – might better correlate with soil moisture over longer temporal scales, such as seasonal or annual (Liang et al., 2024). In regard to our findings, it is important to acknowledge that the optimal temporal scale for estimating daily fluctuations in soil moisture can vary according to soil drainage conditions (Parent et al., 2006) and initial wetness conditions characterizing specific climate zones (Chai et al., 2020) or resulting from different seasonal and annual variations in large-scale climate patterns (Li et al., 2024).

## 425   4.3 Temporal stability of soil moisture patterns

Different initial wetness conditions can also influence the processes controlling spatial variability in soil moisture (Famiglietti et al., 1998; Western et al., 2004; Joshi and Mohanty, 2010; Mei et al., 2018; Gao et al., 2020; Wang et al., 2023). Although



the ranking among predictors remained nearly constant over the study season, we observed that their predictive power changed non-uniformly in relation to daily fluctuations in wetness conditions (i.e., variables responded differently to the same wetness conditions in any day) (Fig. 5). Previous studies indicated that, under drying conditions, lateral water movement is gradually replaced by vertical water movement (Grayson et al., 1997; Western et al., 1999; Rosenbaum et al., 2012), and the spatial variability in soil moisture is likely due to diverse infiltration and evapotranspiration rates related to the spatial distribution of soil and vegetation features (Teuling and Troch, 2005; Takagi and Lin, 2012; Jia et al., 2013; Launiainen et al., 2019). Conversely, the soil moisture spatial variability under rewetting conditions is mostly determined by topographical structures that guide lateral subsurface flow and surface runoff (Grayson et al., 1997; Gaur and Mohanty, 2013). These findings are in line with the results of our study, suggesting that higher infiltration rates in loamy sand soils compared to other soil types increasingly contributed to the observed spatial distribution of soil moisture particularly during drying periods (e.g., end of August to mid-September in our case), while most topographic variables became steadily less relevant during this time. On the other hand, during large precipitation events, topographic indices showed an initial drop in the predictive power likely due to the accumulation of water in the top soil layer and the consequent reduced spatial variability in soil moisture among sites, followed by a time-lagged peak in the predictive power, likely associated with the beginning of lateral subsurface flow driven by topographical features (Grabs et al., 2012). Regarding vegetation, we did not find a direct evidence that evapotranspiration influenced the spatial distribution of soil moisture during short wet-dry transition periods as expected (Teuling et al., 2006). However, we observed a clear seasonal pattern: during the peak of the growing season, generally characterized by warmer and longer days, the spatial heterogeneity of vegetation usually had a larger effect on soil moisture distribution, whereas this influence diminished towards the end of the summer, when days were usually cooler and shorter. Seasonal patterns in solar radiation affected evapotranspiration rates and soil moisture levels differently not only in forests compared to peatlands (Mackay et al., 2007), but also depending on tree species composition, with pine being potentially more responsive to high radiation than spruce (Lagergren and Lindroth, 2002). These findings reiterate the importance of considering the temporal stability of spatial soil moisture patterns under changing wetness conditions (Wang et al., 2023), and we suggest that future research should focus on modeling soil moisture dynamics over longer time scales, beyond a single growing season, particularly in high-latitude environments, where this remains an underexplored topic.

## 5 Conclusions

The Krycklan field infrastructure provided a unique setting for designing a comprehensive study to advance our understanding of the relationship between surface soil moisture and its controls in a forest boreal landscape. By combining remote sensing and modeled data with field measurements across 82 sites in the Krycklan catchment, this study is among the first to examine such a broad range of climatic and environmental factors at different spatial resolutions and temporal scales, focusing on both the spatial and temporal components of soil moisture variability. Our findings suggest that topographical features, soil properties, vegetation characteristics, land use/land cover, and meteorological forcings should all be included when modeling

and mapping variations in soil moisture. We highlight the importance of identifying the optimal spatial resolution and temporal scale for each predictor and considering the dynamic nature of the relationship between soil moisture and its controls, which varies over time.

**Code and data availability**

The code and data used in this study are available from the corresponding author upon request.

**Author contribution**

FZ, AÅ, and WL were responsible for the conceptualization of the study. FZ, CG, JL, and RH conducted fieldwork. WL and JL provided the data for the terrain indices. FZ was responsible for the data processing and analysis, prepared the manuscript including all figures, and led the writing of the paper with contributions from all the co-authors. Funding acquisition AÅ, WL, and CG.

**Competing interests**

The authors declare that they have no conflict of interest.

**Acknowledgements**

We thank the skilled scientists, technicians, and students that have collated the massive amount of data available for the Krycklan catchment.

**Financial support**

This work was funded by The Swedish Research Council Formas (proj no. 2021-00713, 2021-00115 to AÅ and 2021-01993 to CG) and Knut and Alice Wallenberg Foundation (2018.0259 Future Silviculture). This work was partially supported by the Wallenberg AI, Autonomous Systems and Software Program – Humanities and Society (WASP-HS) funded by the Marianne and Marcus Wallenberg Foundation, the Marcus and Amalia Wallenberg Foundation. The funding sources had no involvement 480 in the study design and collection, analysis and interpretation of data, nor in the writing of the report.

**Supplement**

Supplementary materials associated with this article can be found, in the online version, at *link*.



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
