# Peer review of "Controls on spatial and temporal variability of soil moisture across a heterogeneous boreal forest landscape"

_EGUsphere, 2024_

## Author Comment (AC1)

**General comments:**

Abstract:

It was somewhat difficult to get the main points of the article from the abstract. The beginning is very broad and so are the descriptions of what precisely was done. I understand that there were a very large number of variables involved so summarizing all relevant aspects is not feasible but perhaps a bit more precision would help. I also find the reasoning for a boreal forest site a bit lacking, surely there are other reasons to look into soil moisture in these areas other than lack of data?

RC1.1: We appreciate Referee #1's feedback on the abstract and will revise it to improve clarity and precision. We will refine the description of the study's objectives and methods to better convey the main points while keeping it concise. Additionally, we will clarify the rationale for studying soil moisture in boreal forests, highlighting their importance in hydrological and ecological processes beyond just data availability.

Introduction:

I'm not entirely convinced about the main goal of this paper and how it is presented. For one, this very much reads like an empirical modelling paper, but a lot of emphasis in the introduction is placed on understanding the mechanisms and processes driving soil moisture. These would, in my opinion, be better studied with more process-based methods such as mechanistic models or field research focusing on the processes themselves instead of the proxies describing them (such as topography-related indices describing flow patterns in a landscape).

Secondly, I understand the benefit of looking into a multitude of different variables at the same time. However, the good side of being more selective with your variables is that you then have to justify them properly and this is where I think the paper is currently lacking. There are some variables that seem to describe the same thing such as two variables for vegetation biomass and two datasets for soil properties without much justification while some aspects are ignored (such as the topographical variation in radiation). Some spatial variables are tested in multiple resolutions while others are not. This problem also reflects to the results and discussion. As the amount of variables is large and the reasonings behind them a bit unclear, it is challenging to cover and understand all the relevant findings. For example, if one of the goals is not to look at how different datasets of the same variables fare (such as SLU vs. SGU soil data and ERA-5 vs. field data), then why include multiple datasets? This is in my opinion one of the very interesting questions in this type of analysis, yet it is ignored.

RC1.2: We thank Referee #1 for the feedback and for expressing these two concerns regarding the introduction of our manuscript.

    a) As Referee #1 and Referee #2 (RC2.1b) correctly highlight, this study does not aim to directly analyze the mechanisms and processes driving soil moisture. Instead, we use a statistical approach (OPLS) to assess potential controls of soil moisture independently, without accounting for interactions between variables or underlying causal mechanisms. While this method does not establish causality, it provides insights into the relative importance of different factors, which can inform future studies using process-based models. To better align the introduction with

the study's scope, we will revise it to reduce emphasis on mechanisms and processes while highlighting how our results can indirectly contribute to both mechanistic and data-driven modeling efforts.

b) One of the strengths and novel aspects of this paper is the wide range of variables we could analyze, leveraging the extensive field data collected in the Krycklan catchment over the years. Few study areas have access to such a diverse dataset. However, we acknowledge Referee #1's concerns regarding variable selection and justification.
To improve clarity, we propose adding a supplementary section with detailed descriptions of how each variable was calculated or obtained. This will ensure that all relevant information is available in one place rather than being scattered across multiple sources.
We also recognize that some variables may appear redundant, such as the two variables for vegetation biomass or the two variables for soil properties. To address this, we propose adding a figure to the results section comparing the performance of different datasets (e.g., field data vs. remote sensing/modeled data), which will strengthen the paper and clarify the rationale for including multiple datasets of the same variable. This is indeed one very interesting aspect to consider in this type of analysis.
Finally, we agree with Referee #1 and Referee #2 (RC2.9b) that incorporating topographical variation in solar radiation will enhance the study. We will include this variable in the revised manuscript.

Methods:

I again appreciate the multitude of variables but I do not think they are sufficiently covered in the method section. It is not enough cite previous papers without providing almost any explanation of what the variables are and how they have been defined. It makes it nearly impossible for the reader to estimate if your results are reasonable and expected when the reader can't know what was measured without going through various papers, some of them in a foreign language (SLU). Perhaps I missed it, but I'd also like to know the original resolution of the raster datasets.

Would it be possible to provide at least a few maps of the main variables for example in the supplement so that the reader can get a better understanding of the catchment? For example topography, vegetation, land cover and soil type would already provide a lot of very useful information.

RC1.3: We appreciate Referee #1's valuable input regarding the clarity and completeness of the methods section.

a) As explained in RC1.2b, we will add a detailed explanation in the supplement describing each variable and how it was defined. This will ensure that all necessary information is readily available without requiring the reader to consult multiple external sources. As part of the description, we will include details on the original resolution of the raster datasets and the methods used to calculate the topographic indices (see RC2.8).

b) We agree that adding maps illustrating key characteristics of the study area will improve the reader's understanding. Therefore, we will include supplementary maps depicting topography, vegetation, land cover, and soil to provide a clearer spatial context for our analysis.

Discussion:

There is in general throughout the article very little discussion of how the site characteristics influence the results and how well these are applicable outside this study area. I'd also pay a bit more attention to why certain results are as they are and be clear in communicating them. For example, in L420, the longer-term effect of soil temperature is likely due to the fact that soil temperature at those depths (28-100 cm) also varies slowly compared to the top soil temperature. While this is rather obvious, it's maybe good to point it out. Similarly, in L444-446, I would spell out more clearly how vegetation patterns impact soil moisture. This can be for example due to increased transpiration during peak growing season or the impact of shading. Daylengths and their temperatures are not very clear explanations.

Furthermore, there are clearly things that are not measured here, that would influence soil moisture variation, for example the spatial variation of meteorological variables and I do think acknowledging those in the discussion is important.

RC1.4: We thank Referee #1 for the comments and suggestions for improving the discussion. We will clarify more explicitly how site characteristics influence our results and ensure that key findings are clearly communicated. Additionally, we acknowledge that certain factors influencing soil moisture, such as microclimate spatial variability, were not accounted for in this study. We will address these limitations in the revised discussion.

**Specific comments:**

You refer several times to your study period as vegetation period. I'm not familiar with the term so could you define it?

RC1.5: As noted by Referee #2 (RC2.11), the vegetation period, or growing season, in Krycklan begins earlier than our study period (starting July 4th). To avoid confusion and be more accurate, we will replace this term with more generic alternatives such as "study period" or "summer." When greater specificity is needed, we will use "three snow-free months in 2022."

Introduction:

L49: "All potential controls" is a very ambitious term and I'm not entirely sure it is, or can be, achieved with black-box models (or with process-based ones either) considering the interplays of soil moisture with many of its predictive variables, the often massive heterogeneity of soil properties and the need for proxy variables such as topographical indices. While I appreciate the scope of this study, I would perhaps phrase this differently.

RC1.6: This is a good observation. We will replace "all potential controls" with "a broad range of factors/variables" and rephrase this sentence to improve clarity and avoid overstating the scope of the study.

L69-71: In relation to the comment above, this is a much clearer version of the same sentence. However, I'm not sure both of these are needed in the same introduction.

RC1.7: Thanks for the kind words. We will keep this sentence, while we will rephrase the previous one (RC1.6).

L78: I understand that the cited papers don't cover areas outside boreal forests and subarctic tundra, but surely this same thing is true in any cold climate with a seasonal snow cover?

RC1.8: This is a good suggestion. We will replace "In boreal forests and sub-arctic tundra" with "In cold-climate regions with seasonal snow cover."

L83: "However, recent research indicates that topography may have a different relationship with soil moisture under varying wetness conditions." This is a rather vague sentence. Do you mean to say that the impact of topography differs depending on the wetness conditions?

RC1.9: We thank Referee #1 and Referee #2 (see RC2.5) for noting that this sentence may not be clear. While many studies indicate that the relationship between topography and soil moisture is strongest during wet periods and weakens as conditions become drier (L77-83), recent research suggests that this is not always the case (L83-84). Some studies report the opposite pattern, where topography plays a more significant role during dry periods than in wet periods (L84-86). To clarify this point, we will rephrase the sentence as follows: "However, recent research suggests that the influence of topography on soil moisture does not always follow this pattern." Alternatively, we may integrate this sentence with the following one for better readability.

Methods:

L116: Is the catchment area primarily managed boreal forest and if yes, how is it managed? I could imagine that managed boreal forests differ in their soil moisture controls compared to non-managed forests so this could at least be mentioned somewhere.

RC1.10: We agree that forest management can indeed influence soil moisture. While approximately 25% of the forest area in the Krycklan catchment has been protected since 1922, the majority consists of second-growth forest, with clear-cutting being the primary management practice. This has resulted in a diverse mosaic of forest stands with varying age classes and species compositions. Although forest management is not explicitly included as a variable in this study, we incorporated several variables reflecting forest status, such as species composition, forest productivity, and tree structure, all of which are intrinsically linked to forest management. Additionally, we included "clear-cut" as a specific class within the "land use" variable. As suggested, we will clarify the management history of the catchment by adding a sentence at the end of the first paragraph in the study area section.

L132: I fully understand the separation of the variables into spatially and temporally varying ones. However, it would be good to somewhere, for example in the discussion, recognize that many of the temporal variables are indeed not spatially homogeneous.

For example air temperature, particularly close to the ground, can vary considerably (and is often tightly connected to soil moisture), transpiration naturally depends on vegetation, radiation on the topography, etc.

RC1.11: Thanks for this valuable observation. We agree that many temporal variables are not spatially homogeneous and that factors such as air and soil temperature can exhibit significant spatial variability due to influences like topography and vegetation. In the discussion, we will acknowledge this limitation and clarify that microclimate spatial variability was not explicitly accounted for in our analysis.

L135: How was the subset selected?

RC1.12: We appreciate Referee #1's comment, which aligns with Referee #2's observation (RC2.6). While we did not adopt a specific formal sampling strategy (e.g., random, stratified, or cluster sampling) for selecting the datalogger locations, we followed key principles outlined in the first paragraph (L137-141). Our main goal was to capture a broad range of soil moisture conditions representative of the variability observed in Swedish forests. To inform our selection, we drew on previous research analyzing field-determined soil moisture classes across Sweden using nearly 20,000 National Forest Inventory (NFI) plots (Figure 3 in Ågren et al. 2021, see graphic below). In Krycklan, we surveyed 500 plots following the same NFI protocol and classified them into five soil moisture classes. We selected 82 plots and instrumented them with dataloggers, ensuring they reflected the NFI soil moisture distribution. Given the high heterogeneity in central Krycklan, we concentrated approximately half of the loggers there. Additionally, some loggers were positioned near permanent measurement stations to facilitate comparisons between different soil moisture measurement systems. The remaining loggers were distributed across the catchment to ensure adequate spatial coverage while maintaining ease of accessibility. We will revise the paragraph to clarify and provide more detail on the selection process.

[Figure]

**Percentages of field plots in the soil moisture classes of the National Forest Inventory (NFI) dataset (n = 19,643). Source: Figure 3 in Ågren et al. (2021). https://doi.org/10.1016/j.geoderma.2021.115280**

L150: Nothing to correct here, just wanted to say well done for adequately explaining how you did the calibration!

RC1.13: We are glad to hear that the explanation of the calibration process was clear and well-received.

Results:

L239: It might be worth noting that the sharp decline during precipitation events starts happening after in August. In July the responses are very small. I would also perhaps use the term "precipitation event" instead of "precipitation occurrence."

RC1.14: This is a good observation. We agree that it is important to highlight the timing of the sharp decline in soil moisture standard deviation during precipitation events, which are more evident in August and September. We will update the manuscript to include this clarification. Additionally, we will replace the term "precipitation occurrence" with "precipitation event" as suggested.

L278: This is very nit-picky, but could you place the abbreviations of plan curvature and downslope index other way around so they're consistent with the rest of the sentence?

RC1.15: Thanks for catching this mistake. We agree that the order of the abbreviations should be consistent with the rest of the sentence. We will revise the manuscript accordingly to ensure consistency.

L281: This is a good example of why explaining the variables in more detail and justifying the selection would be beneficial. Now it very much seems that you're trying to explain soil moisture by examining soil moisture, while it probably is just interesting to see how well these two correspond with each other. The same goes for the pine variables in the next paragraph (L289 and L290).

RC1.16: We agree with Referee #1's comment. Indeed, our intention was to compare the performance of the same variable from different datasets. As mentioned in RC1.2b, we will include a more detailed description of the variables in the supplementary material and justify our choice of using different datasets for the same variable by comparing their performance in a new, dedicated figure.

L294: The end of the sentence is missing something.

RC1.17: We appreciate Referee #1's comment. Upon reviewing the sentence, we believe it is complete. The structure of the sentence was intended to contrast the performance of the predictors in terms of their predictive power (predictive axis) versus the noise they introduce into the model (orthogonal axis).

Discussion:

L373: I'd be careful when using the word predict. In my understanding, this type of modelling is trying to explain the variation, since there aren't predictions outside the measurements.

RC1.18: We understand the concern. OPLS is inherently a predictive modeling technique, but in our study, we primarily use it to explain the variation in soil moisture within the observed dataset rather than to predict values for independent data. While "predict" is a standard term in multivariate regression modeling, we acknowledge that "explain" may be more appropriate in this context. To ensure clarity, we will use "explain" in L373 and other instances where "predict" could be misleading, while retaining "predicting" or "predictive" when specifically referring to OPLS model characteristics, components, and results.

L395: I'm not entirely convinced that Kemppinen's study site is all that comparable with the Krycklan catchment considering the difference in vegetation (treeless tundra vs. boreal forest) but it's also very difficult to say since there are little maps providing information on the characteristics of your cathcment. As an interesting side note, having visited the valley, I'd suspect that the reason for TWI being more useful there is due to the shape of the valley which very strongly gathers the water flow to low-lying areas (and there are also deep organic layers at the bottom of the valley due to this, further enhancing the accumulation of soil moisture). This is probably a good example of exactly what you also show in the paper, that the characteristics of different watersheds are important.

RC1.19: This aligns with a comment from Referee #2 (RC2.13). We agree that the differences in vegetation between the study sites in the Krycklan catchment and Kemppinen's site (boreal forest vs. treeless tundra) may limit the comparability of the two landscapes. Additionally, as the reviewer noted, the topographic characteristics and valley shape in Kemppinen's study area could play a key role in the relevance of the topographic wetness index (TWI) in that landscape. We will revise the sentence accordingly to reflect these considerations (L393-397).

L399: "...their spatial resolutions and thresholds. *We* argue that…"

RC1.20: As suggested, we will separate this sentence into two sentences.

L420: It might be useful to point out that such a deep soil temperature also fluctuates very slowly compared to for example top soil temperature. Furthermore, finer spatial resolution of air temperature might have yielded different results.

RC1.21: We thank Referee #1 for pointing this out. We agree that deep soil temperature fluctuates more slowly compared to topsoil temperature. However, since soil temperature in the upper layers of the soil showed no correlation with soil moisture in our study (Fig. S2), we believe it may not be necessary to emphasize this here. On the other hand, we agree that a higher spatial resolution of air and soil temperature could have yielded different results. As discussed in RC1.4, we will include this consideration in the discussion.

L442: "Regarding vegetation, we did not find a direct evidence..."

RC1.22: We are unsure if we fully understood Referee #1's comment or the suggested changes. Could you please clarify what aspects we should address or modify in this sentence, as it appears the same in the manuscript?

L443: I'm not sure that the article by Teuling et al is very useful here. First of all, it looked at evapotranspiration driven by soil moisture, not the other way around. This is somewhat semantics but I do think it's good to remember which processes drive which (or if they are driving one another). Secondly, and this would be interesting to study further, the Teuling-article studied single points in various ecosystems whereas you're concentrating on much more fine-scale variation of soil moisture. These might not behave in a similar way.

RC1.23: As suggested, we will remove this reference here.

L447: Could you be a bit more precise here with the word "differently"?

RC1.24: Seasonal patterns in solar radiation affect evapotranspiration rates and soil moisture levels in forests and peatlands in distinct ways. In forests, evapotranspiration tends to be higher, more variable, and more responsive to changes in solar radiation due to canopy cover. In contrast, peatlands typically exhibit more stable soil moisture levels and lower evapotranspiration rates due to waterlogged conditions and lower biomass. We will include a concise addition to the text to provide further clarity.

Figures:

Fig. 1: Ignore this comment, if you think it's not suitable, but would it be possible to get the main streams within the catchment area visible on the map?

RC1.25: This is a good suggestion. We will update Fig. 1 to include the main streams within the catchment area for reference.

Fig. 2: Overall an informative figure, but could perhaps the arrows on "no trend" sites be removed for clarity? Also, in the legend of 2c, the symbol of ERA5-Land is indistinguishable from the other lines, it might help making the lines in the legend somewhat thicker than the actual lines in the plot.¨

RC1.26: As suggested, we will remove the arrows for the sites with no trend in Fig. 2b for better clarity, and we will make the lines in the legend of Fig. 2c thicker to improve distinction between the symbols.

Fig. 3: Should the resolution be in meters or in square meters?

RC1.27: We thank Referee #1 for the observation. The spatial resolution throughout the text is expressed in meters. It represents the length of the cell side in the gridded datasets, rather than the area. For consistency, we would like to maintain this unit (meters) in the figure as well.

---

## Author Comment (AC2)

**General comments:**

In my opinion, the most concrete contribution that the study can make is to identify and provide key spatial and temporal predictors for data-driven models used in soil moisture mapping. While this aim and contribution is relevant and briefly mentioned, it would benefit from being more explicitly emphasized and clarified throughout the manuscript, particularly in the abstract, introduction, and discussion. Strengthening these sections could better position the study within the broader context of soil moisture research and mapping.

It remains unclear whether the study contributes to understanding the processes driving soil moisture variability. Therefore, it is also doubtful whether the study offers a contribution to the development of physically based models. Such models already incorporate key spatial (e.g., vegetation, soil texture, topography) and temporal (e.g., meteorological forcing) factors, and the manuscript does not address how the findings could enhance these models. Perhaps the authors could emphasize the need for more accurate field and remote sensing data on the identified variables, which could indirectly benefit physically based model approaches that rely on such data.

Although this study may not directly advance our understanding of the processes driving soil moisture variability, the identification of key predictors, many of which are already incorporated in physically based models, presents an opportunity for the authors to discuss how these different methods (process-based and data-driven) could be integrated for soil moisture mapping (e.g. hybrid approaches).

The amount of predictors and scales is impressive, but there are still many limitations to this study (as to any study). It is not a major problem, but the limitations need more attention in the discussion. For instance, topographic and tree shading is neglected, the spatial scales go only up to 30 or 64 m, and temporal extend was only a few months, just to name a few.

RC2.1: We thank Referee #2 for these valuable suggestions.

   a) We will more clearly emphasize the aim and concrete contribution of this paper in the abstract, introduction, and discussion to better position our study within soil moisture research and mapping.
   b) We acknowledge that our study does not focus on the mechanisms driving soil moisture but rather on identifying key spatial and temporal predictors for data-driven models. As noted in RC1.2a, we will reduce the emphasis on processes and provide concrete examples (e.g., enhanced soil maps) of how our findings could benefit both data-driven and physically based approaches.
   c) This is an interesting point. We will expand the discussion to explore potential synergies between data-driven and process-based methods in the context of soil moisture mapping.
   d) We will discuss the key study limitations, including the omission of microclimate spatial predictors (RC1.4, RC1.11) and the restriction of our analysis to only one incomplete snow-free season (RC2.4). Additionally, we will directly address the lack of topographic shading by incorporating it into the analysis (see RC1.2b, RC2.9b).

**Specific comments:**

L33-34: Please add reference to "influence soil nitrogen availability"

RC2.2: The reference to Nogovitcyn et al. (2023), which is already cited in the main text, covers both "soil nitrogen availability" and "needle production".

L36-39: It would help to have example reference to each of these applications that require soil moisture state.

RC2.3: Thanks for the observation. We have already included references for all the mentioned applications at the end of the sentence (L38-39), with each reference covering one or more applications. To improve clarity, we can either add more references or explicitly match each reference to its corresponding application.

L77-80: This is a good point that topography can have a major role in the early summer. In this study, you basically skipped the period impacted by snowmelt. How do you think the results would differ if full snow-free season was included?

RC2.4: As suggested, we will address this point when discussing the key study limitations (see RC2.1d). Indeed, topography is likely to play a more significant role after snowmelt in our study area as well.

L83-84: I don't quite understand this sentence. You already described how the impact of topography changes between wet and dry seasons. Perhaps you mean that this may change within the season as well. This could be reformulated. Also please add references to the recent research you're referring to.

RC2.5: As both Referee #2 and Referee #1 (see RC1.9) highlight, this sentence is unclear. Most studies suggest that topography has a stronger influence on soil moisture during wet periods and a weaker influence during dry periods (L77-83). However, recent research challenges these findings, showing cases where topographic metrics remain equally strong or even become more important during dry periods, or where their influence stays consistently low during wet periods (L83-86). To improve clarity, we will rephrase the sentence as follows: "However, recent research suggests that the influence of topography on soil moisture does not always follow this pattern." Alternatively, we may integrate this sentence with the following one for better readability. The references for the recent research we are referring to are already included (L85-86).

L135: How were the soil moisture locations decided? They mostly fall in the intensitve monitoring area. How much of them are on peat soils? How much in mineral? Why this subset of the measurements?

RC2.6: We appreciate Referee #2's comment. As noted in RC1.12, we did not use a formal sampling strategy (e.g., random or stratified sampling) but followed key principles outlined in L137-141. Our primary goal was to mirror the distribution of soil moisture conditions representative of Swedish forests (please see RC1.12 for a more detailed explanation and a graphic visualization). To achieve this, we placed about half of the loggers in the highly heterogeneous central part of the Krycklan catchment. Some loggers were positioned near permanent stations for comparison between different

measurement systems, while the rest were distributed to balance spatial coverage and accessibility. We will revise the manuscript to clarify the selection process and add details on the proportion of loggers in peat versus mineral soils. This will also be visible in new supplementary figures showing land cover and soil information in relation to logger locations. (see RC1.3b and RC2.9).

L137: I was surprised not to see any fully saturated measurements on peat soils (peat soil porosity is around 0.90). Were there no peatlands (or peatlands with measurements) where the water table was near the surface? Hence, I don't think these measurements "capture the full spectrum of soil moisture levels across the Swedish landscape."

RC2.7: Thanks for this observation. Our loggers were placed across a wide range of soil moisture conditions, including very wet peat soils with water tables near the surface. However, our measurements of volumetric water content (VWC) range from 0 to 0.60 rather than 0 to 1 because VWC accounts for the total volume of soil, including solid particles, water, and air. Even in fully saturated peat, VWC does not reach 100% because soil particles occupy a portion of the total volume. The observed maximum values (~60%) are consistent with expectations for saturated peat soils, considering their high porosity (~90%) and the inherent structure of organic material. To clarify this point, we will refine our description in the methods section and/or in the caption of Figure 2.

L167.: While citing for details is understandable, it would be good to elaborate a bit more on how the topographic indices were defined. For example, it makes a difference which method is used (e.g. D8 vs. Dinf). Please describe also the site quality index.

RC2.8: We thank Referee #2 for pointing this out. As the definition of variables – not only the topographic indices but also all other predictors – is a major shared concern with Referee #1 (see RC1.2b and RC1.3a), we will provide a detailed description of each variable in the supplement, including the methods used to calculate the topographic indices. This will ensure transparency and accuracy, allowing readers to interpret the results correctly.

Sect. 2.2.2: Please consider adding spatial predictor figure in supplement. It would give the reader a better idea how the landscape looks like. It is an impressive number of predictors, but one obvious one is missing: topographic shading.

RC2.9: Thanks for the suggestions.

   a) We agree that adding maps illustrating some of the predictors would provide a better spatial context for the study landscape. As suggested by Referee #1, we will include supplementary maps of topography, vegetation, land cover, and soil (see RC1.3b).
   b) Both Referee #2 and Referee #1 (RC1.2b) identified topographical variation in solar radiation, including the effect of topographic shading, as an important variable that is currently missing. Therefore, we will rerun the analysis with this additional variable and revise the manuscript accordingly.

Sect. 3.2. repeats many of the things already described in the methods (Sect. 2.3., which is great by they way). Perhaps the chapter in 3.2. can be integrated to 2.3.?

RC2.10: Thanks for this comment. In Section 2.3, we provide the statistical background of the OPLS analysis, describe the different models created, and outline the metrics used for evaluation. In Section 3.2, however, our focus is on the visualization aspect, helping readers interpret the figures by explaining plot elements such as shapes, colors, and loading positions. This prevents repetitions when describing the results in the following subsections and is particularly useful for readers unfamiliar with the method. We believe that placing this explanation at the beginning of the results section, where the OPLS plots are presented, enhances readability. However, we acknowledge the reviewer's concern and will refine the text to improve conciseness and avoid redundancy between sections 2.3. and 3.2.

L349: What is a vegetation period? Analysis starts from July something, growing season starts earlier.

RC2.11: This is a good point. Indeed, the growing season starts earlier in Krycklan, making the term not entirely appropriate in this case. As mentioned in RC1.5, we will instead use broader terms like "study period" or "summer." When more detail is needed, we will specify "three snow-free months in 2022."

L370: "forest boreal landscape" -> "boreal forest landscape"

RC2.12: Thanks for identifying this mistake. We will make the correction accordingly.

L394: Kemppinen et al. study site (tundra) is quite different to Krycklan.

RC2.13: We thank Referee #2 for the comment, which echoes a remark from Referee #1 (RC1.19). As noted earlier, we will revise this sentence to avoid a direct comparison between the two study areas (L393-397).

L420: Consider replacing "performed poorly" with "correlated poorly with soil moisture" or something similar.

RC2.14: As suggested, we will replace "performed poorly" with "showed weak correlation with soil moisture."

Code and data availability: Better practice would be to share the data in an openly available repository. Or has the data already been shared in the cited literature?

RC2.15: We appreciate this suggestion. The data for nearly all variables used in the analysis have already been made publicly available in the cited literature. The only exceptions are the field data from SLU (2021) and the soil moisture data from the TOMST loggers. We will create a data repository for these two datasets and update the data availability statement accordingly.

---

## Author Response (AR1)

**REFEREE #1**

The authors have examined top soil moisture and the myriad of environmental and climatological variables controlling it in a well-studied catchment area in northern Sweden. The study design includes a good amount of top soil moisture measurements and studies the impact of different variables, as well as their resolution/calculation methods, on the spatial and temporal variation of soil moisture. Overall, the manuscript is well executed and the study design interesting. However, there are a few smaller and bigger aspects of the paper that would benefit from some additional work.

We sincerely thank Referee #1 for summarizing the key elements of the paper, providing overall positive feedback, and offering constructive criticism and suggestions, which have strengthen the manuscript. Reviewer comments are shown in black font and author responses in blue font, with responses numbered R1.1–R1.27 for clarity. Changes are also highlighted in light blue in the manuscript and supplement.

**General comments:**

**Abstract:**

It was somewhat difficult to get the main points of the article from the abstract. The beginning is very broad and so are the descriptions of what precisely was done. I understand that there were a very large number of variables involved so summarizing all relevant aspects is not feasible but perhaps a bit more precision would help. I also find the reasoning for a boreal forest site a bit lacking, surely there are other reasons to look into soil moisture in these areas other than lack of data?

R1.1: We appreciate Referee #1's feedback on the abstract and we revised it to improve clarity and precision. We refined the description of the study's objectives and methods to better convey the main points while keeping it concise (L5-16). Additionally, we clarified the rationale for studying soil moisture in boreal forests, highlighting their importance in hydrological and ecological processes in the context of climate change beyond just data availability (L3-4).

**Introduction:**

I'm not entirely convinced about the main goal of this paper and how it is presented. For one, this very much reads like an empirical modelling paper, but a lot of emphasis in the introduction is placed on understanding the mechanisms and processes driving soil moisture. These would, in my opinion, be better studied with more process-based methods such as mechanistic models or field research focusing on the processes themselves instead of the proxies describing them (such as topography-related indices describing flow patterns in a landscape).

Secondly, I understand the benefit of looking into a multitude of different variables at the same time. However, the good side of being more selective with your variables is that you then have to justify them properly and this is where I think the paper is currently lacking. There are some variables that seem to describe the same thing such as two variables for vegetation biomass and two datasets for soil properties without much

justification while some aspects are ignored (such as the topographical variation in radiation). Some spatial variables are tested in multiple resolutions while others are not. This problem also reflects to the results and discussion. As the amount of variables is large and the reasoning behind them a bit unclear, it is challenging to cover and understand all the relevant findings. For example, if one of the goals is not to look at how different datasets of the same variables fare (such as SLU vs. SGU soil data and ERA-5 vs. field data), then why include multiple datasets? This is in my opinion one of the very interesting questions in this type of analysis, yet it is ignored.

**R1.2: We thank Referee #1 for the feedback and for expressing these two concerns regarding the introduction of our manuscript.**

- a) As Referee #1 and Referee #2 (R2.1b) correctly highlight, this study does not aim to directly analyze the mechanisms and processes driving soil moisture. Instead, we used a statistical approach (OPLS) to assess potential controls of soil moisture independently, without accounting for interactions between variables or underlying causal mechanisms. While this method does not establish causality, it provides insights into the relative importance of different factors, which can inform future studies using data-driven models. To better align the introduction with the study's scope, we revised it to reduce emphasis on mechanisms and processes while highlighting how our results can indirectly contribute to data-driven modeling efforts (L46, L53-56, L73-77, and L118).
- b) One of the strengths and novel aspects of this paper is the wide range of variables we could analyze, leveraging the extensive field data collected in the Krycklan catchment over the years. Few study areas have access to such a diverse dataset. However, we acknowledge Referee #1's concerns regarding variable selection and justification. To improve clarity, we added a supplementary section with detailed descriptions of how each variable was calculated or obtained. This ensures that all relevant information is available in one place rather than being scattered across multiple sources. We revised the text accordingly (L199-200, L209, L222, and L228).

We acknowledge that some variables may appear redundant, such as the two variables for vegetation biomass or the two variables for soil properties. To address this concern, we introduced a new specific research aim (aim iv) (L122-123), addressed to compare field data vs. remote sensing/modeled data, including analogous variables from these two different sources. We restructured the paper accordingly, with a dedicated section in the abstract (L27-29), introduction (L51-53, L96-108, L113-114), methods (L251-L257), results (L408-421, including a new figure (Fig. 6)), and discussion (L515-545). We compared the performance of different datasets (e.g., field data vs. remote sensing/modeled data), thereby clarifying the rationale for including multiple datasets representing the same variable. Although this analysis was part of the original research plan, it was initially omitted to meet length constraints. We appreciate the reviewer's recognition of the importance of this aspect, and we agree that it adds valuable depth to the overall analysis.

Finally, we agree with Referee #1 and Referee #2 (R2-9b) that topographical variation in solar radiation is an important and easy-to-calculate variable that was missing in the previous version of the manuscript. Accordingly, we rerun the spatial OPLS analysis, now including both diffuse solar radiation and direct solar

radiation derived from the original DEM. We decided to exclude slope and aspect because the information they carried is inherently captured in the calculation of the solar radiation variables (see detailed description in the Supplement). Since this analysis had to be rerun, we also took the opportunity to include two additional variables that were previously missing: (1) the clear-cut class from the Land map, to match the analogous variable determined in the field, and (2) the glacifluvial sediment class from the SGU Quaternary deposit map, due to the inclusion of an additional layer of information (see detailed description in the Supplement). As a result, we updated Table 1, Figures 3, 5, S2, and S4, as well as the corresponding text in the manuscript (e.g., L318-319)."

**Methods:**

I again appreciate the multitude of variables but I do not think they are sufficiently covered in the method section. It is not enough cite previous papers without providing almost any explanation of what the variables are and how they have been defined. It makes it nearly impossible for the reader to estimate if your results are reasonable and expected when the reader can't know what was measured without going through various papers, some of them in a foreign language (SLU). Perhaps I missed it, but I'd also like to know the original resolution of the raster datasets.

Would it be possible to provide at least a few maps of the main variables for example in the supplement so that the reader can get a better understanding of the catchment? For example topography, vegetation, land cover and soil type would already provide a lot of very useful information.

R1.3: We appreciate Referee #1's valuable input regarding the clarity and completeness of the methods section.

- a) As explained in R1.2b, we added a detailed explanation in the supplement describing each variable and how it was defined. This ensures that all necessary information is readily available without requiring the reader to consult multiple external sources. As part of the description, we included details on the original resolution of the raster datasets and the methods used to calculate the topographic indices (see R2.8).
- b) We agree that adding maps illustrating key characteristics of the study area will improve the reader's understanding. Therefore, we added in the supplementary material a 4-panel map (Fig. S1) with information regarding soil (quaternary deposits, Fig. S1a), topography (elevation, Fig. S1b), vegetation (normalized difference vegetation index, Fig. S1c), and land use/land cover (Fig. S1d). This map provides a clearer spatial context for our analysis.

**Discussion:**

There is in general throughout the article very little discussion of how the site characteristics influence the results and how well these are applicable outside this study area. I'd also pay a bit more attention to why certain results are as they are and be clear in communicating them. For example, in L420, the longer-term effect of soil temperature is likely due to the fact that soil temperature at those depths (28-100 cm) also varies slowly compared to the top soil temperature. While this is rather obvious, it's maybe good to point it out. Similarly, in L444-446, I would spell out more clearly how vegetation

patterns impact soil moisture. This can be for example due to increased transpiration during peak growing season or the impact of shading. Daylengths and their temperatures are not very clear explanations.

Furthermore, there are clearly things that are not measured here, that would influence soil moisture variation, for example the spatial variation of meteorological variables and I do think acknowledging those in the discussion is important.

R1.4: We thank Referee #1 for the comments. As suggested, we have now specified that deep soil temperature fluctuates more slowly compared to topsoil temperature (L479). We have also clarified how vegetation impacted soil moisture (L506-507 and L509). Additionally, we acknowledge that certain factors influencing soil moisture, such as microclimate spatial variability, were not accounted for in this study. We addresses these limitations in the revised discussion (L527-530).

**Specific comments:**

You refer several times to your study period as vegetation period. I'm not familiar with the term so could you define it?

R1.5: As noted by Referee #2 (R2.11), the vegetation period, or growing season, in Krycklan begins earlier than our study period (starting July 4). To avoid confusion and be more accurate, we have now replaced this term with more generic terms such as "study period" (L388) and "summer" (L17), or more specific alternatives, such as "three snow-free months in 2022" (L110 and L434).

**Introduction:**

L49: "All potential controls" is a very ambitious term and I'm not entirely sure it is, or can be, achieved with black-box models (or with process-based ones either) considering the interplays of soil moisture with many of its predictive variables, the often massive heterogeneity of soil properties and the need for proxy variables such as topographical indices. While I appreciate the scope of this study, I would perhaps phrase this differently.

R1.6: This is a good observation. We deleted "all potential controls" as part of the process of rephrasing this sentence to clarify the scope of the study (L53-56).

L69-71: In relation to the comment above, this is a much clearer version of the same sentence. However, I'm not sure both of these are needed in the same introduction.

R1.7: Thanks for the kind words. We agree with the reviewer that this sentence and the previous one (R1.6) were redundant, so we rephrased both of them. The first one (L53-56) now clarifies the overall scope of the study as it concludes the first paragraph of the introduction, while this one (L73-77) refers specifically to the content of the second paragraph.

L78: I understand that the cited papers don't cover areas outside boreal forests and subarctic tundra, but surely this same thing is true in any cold climate with a seasonal snow cover?

R1.8: This is a good suggestion. We replaced "In boreal forests and sub-arctic tundra" with "In cold-climate regions with seasonal snow cover" (L83-84).

L83: "However, recent research indicates that topography may have a different relationship with soil moisture under varying wetness conditions." This is a rather vague sentence. Do you mean to say that the impact of topography differs depending on the wetness conditions?

R1.9: We thank Referee #1 and Referee #2 (see R2.5) for highlighting that this sentence may not be clear. While many studies indicate that the relationship between topography and soil moisture is strongest during wet periods and weakens as conditions become drier (L83-89), recent research suggests that this is not always the case (L89-90). Some studies report the opposite pattern, where topography plays a more significant role during dry periods than in wet periods (L90-92). To clarify this point and for better readability, we rephrased two sentences (L89-92).

**Methods:**

L116: Is the catchment area primarily managed boreal forest and if yes, how is it managed? I could imagine that managed boreal forests differ in their soil moisture controls compared to non-managed forests so this could at least be mentioned somewhere.

R1.10: We agree that forest management can indeed influence soil moisture. While approximately 25% of the forest area in the Krycklan catchment has been protected since 1922, the majority consists of second-growth forest, with clear-cutting being the primary management practice. This has resulted in a diverse mosaic of forest stands with varying age classes and species compositions. Although forest management is not explicitly included as a variable in this study, we incorporated several variables reflecting forest status, such as species composition, forest productivity, and tree structure, all of which are intrinsically linked to forest management. Additionally, we included "clear-cut" as a specific class within the "land use" variable. As suggested, we clarified the management history of the catchment by adding two sentences at the end of the first paragraph in the study area section (L136-140).

L132: I fully understand the separation of the variables into spatially and temporally varying ones. However, it would be good to somewhere, for example in the discussion, recognize that many of the temporal variables are indeed not spatially homogeneous. For example air temperature, particularly close to the ground, can vary considerably (and is often tightly connected to soil moisture), transpiration naturally depends on vegetation, radiation on the topography, etc.

R1.11: Thanks for this valuable observation. We agree that many temporal variables are not spatially homogeneous and that factors such as air and soil temperature can exhibit significant spatial variability due to influences like topography and vegetation. In the discussion, we acknowledged this limitation and clarified that microclimate spatial variability was not explicitly accounted for in our analysis (L527-530).

R1.12: We appreciate Referee #1's comment, which aligns with Referee #2's observation (R2.6). While we did not adopt a specific formal sampling strategy (e.g., random. stratified, or cluster sampling) for selecting the datalogger locations, we followed key principles outlined in the first paragraph. Our main goal was to capture a broad range of soil moisture conditions representative of the variability observed in Swedish forests. To inform our selection, we drew on previous research analyzing field-determined soil moisture classes across Sweden using nearly 20,000 National Forest Inventory (NFI) plots (Figure 3 in Ågren et al. 2021, see graphic below). In Krycklan, we surveyed 500 plots following the same NFI protocol and classified them into five soil moisture classes. We selected 82 plots and instrumented them with dataloggers, ensuring they reflected the NFI soil moisture distribution. Given the high heterogeneity in central Krycklan, we concentrated approximately half of the loggers there. Additionally, some loggers were positioned near permanent measurement stations to facilitate comparisons between different soil moisture measurement systems. The remaining loggers were distributed across the catchment to ensure adequate spatial coverage while maintaining ease of accessibility. We have now revised the paragraph to clarify and provide more detail on the selection process (L159-167).

Percentages of field plots in the soil moisture classes of the National Forest Inventory (NFI) dataset (n = 19,643). Source: Figure 3 in Ågren et al. (2021). <a href="https://doi.org/10.1016/j.geoderma.2021.115280">https://doi.org/10.1016/j.geoderma.2021.115280</a>

L150: Nothing to correct here, just wanted to say well done for adequately explaining how you did the calibration!

R1.13: We are glad to hear that the explanation of the calibration process was clear and well-received.

**Results:**

L239: It might be worth noting that the sharp decline during precipitation events starts happening after in August. In July the responses are very small. I would also perhaps use the term "precipitation event" instead of "precipitation occurrence."

R1.14: This is a good observation. We agree that it is important to highlight the timing of the sharp decline in soil moisture standard deviation during precipitation events, which are more evident in August and September. We updated the manuscript to include this clarification (L278-279). Additionally, we replaced the term "precipitation occurrence" with "precipitation event" as suggested (L278).

L278: This is very nit-picky, but could you place the abbreviations of plan curvature and downslope index other way around so they're consistent with the rest of the sentence?

R1.15: Thanks for catching this mistake. We agree that the order of the abbreviations should be consistent with the rest of the sentence. We revised the manuscript accordingly to ensure consistency (L316).

L281: This is a good example of why explaining the variables in more detail and justifying the selection would be beneficial. Now it very much seems that you're trying to explain soil moisture by examining soil moisture, while it probably is just interesting to see how well these two correspond with each other. The same goes for the pine variables in the next paragraph (L289 and L290).

R1.16: We agree with Referee #1's comment. Indeed, our intention was to compare the performance of the same variable from different datasets. As mentioned in R1.2b, we included a detailed description of all the variables in the supplementary material and justify our choice of using different datasets for the same variable by introducing a new study aim with a new dedicated figure (Fig. 6) (see R2.1b).

L294: The end of the sentence is missing something.

R1.17: We appreciate Referee #1's comment. Upon reviewing the sentence, we believe it is complete. The structure of the sentence was intended to contrast the performance of the predictors in terms of their predictive power (predictive axis) versus the noise they introduce into the model (orthogonal axis) (L332-333).

**Discussion:**

L373: I'd be careful when using the word predict. In my understanding, this type of modelling is trying to explain the variation, since there aren't predictions outside the measurements.

R1.18: We understand the concern. OPLS is inherently a predictive modeling technique, but in our study, we primarily use it to explain the variation in soil moisture within the observed dataset rather than to predict values for independent data. While "predict" is a standard term in multivariate regression modeling, we acknowledge that "explain" may be more appropriate in this context. To ensure clarity, we have now used "explain" in L434 and other instances where "predict" could be misleading, while retaining

"predicting" or "predictive" when specifically referring to OPLS model characteristics, components, and results.

L395: I'm not entirely convinced that Kemppinen's study site is all that comparable with the Krycklan catchment considering the difference in vegetation (treeless tundra vs. boreal forest) but it's also very difficult to say since there are little maps providing information on the characteristics of your cathcment. As an interesting side note, having visited the valley, I'd suspect that the reason for TWI being more useful there is due to the shape of the valley which very strongly gathers the water flow to low-lying areas (and there are also deep organic layers at the bottom of the valley due to this, further enhancing the accumulation of soil moisture). This is probably a good example of exactly what you also show in the paper, that the characteristics of different watersheds are important.

R1.19: This aligns with a comment from Referee #2 (R2.13). We agree that the differences in vegetation between the study sites in the Krycklan catchment and Kemppinen's site (boreal forest vs. treeless tundra) may limit the comparability of the two landscapes. Additionally, as the reviewer noted, the topographic characteristics and valley shape in Kemppinen's study area could play a key role in the relevance of the topographic wetness index (TWI) in that landscape. We have now deleted this comparison to reflect these considerations, and we added this reference at the beginning of the paragraph as an example of modeling soil moisture through LiDAR-derived topographic indices (L440).

L399: "...their spatial resolutions and thresholds. We argue that..."

R1.20: As suggested, we split this sentence into three sentences (L457-460).

L420: It might be useful to point out that such a deep soil temperature also fluctuates very slowly compared to for example top soil temperature. Furthermore, finer spatial resolution of air temperature might have yielded different results.

R1.21: We thank Referee #1 for pointing this out. As mentioned in R1.4, we have now specified that deep soil temperature fluctuates more slowly compared to topsoil temperature (L479). We also agree that a higher spatial resolution of air and soil temperature could have yielded different results. As discussed in R1.4, we have now included this consideration in the discussion (L527-530).

L442: "Regarding vegetation, we did not find a direct evidence..."

**R1.22: We removed this sentence.**

L443: I'm not sure that the article by Teuling et al is very useful here. First of all, it looked at evapotranspiration driven by soil moisture, not the other way around. This is somewhat semantics but I do think it's good to remember which processes drive which (or if they are driving one another). Secondly, and this would be interesting to study further, the Teuling-article studied single points in various ecosystems whereas you're concentrating on much more fine-scale variation of soil moisture. These might not behave in a similar way.

- R1.23: As suggested, we have now removed this reference.
- L447: Could you be a bit more precise here with the word "differently"?
- R1.24: Seasonal patterns in solar radiation affect evapotranspiration rates and soil moisture levels in forests and peatlands in distinct ways. In forests, evapotranspiration tends to be higher, more variable, and more responsive to changes in solar radiation due to canopy cover. In contrast, peatlands typically exhibit more stable soil moisture levels and lower evapotranspiration rates due to waterlogged conditions and lower biomass. We included a concise addition to the text to provide further clarity (L509).

**Figures:**

- Fig. 1: Ignore this comment, if you think it's not suitable, but would it be possible to get the main streams within the catchment area visible on the map?
- R1.25: This is a good suggestion. We have now included the main streams and lakes within the catchment area in Fig. 1 for better reference. The figure legend and caption have been updated accordingly.
- Fig. 2: Overall an informative figure, but could perhaps the arrows on "no trend" sites be removed for clarity? Also, in the legend of 2c, the symbol of ERA5-Land is indistinguishable from the other lines, it might help making the lines in the legend somewhat thicker than the actual lines in the plot.
- R1.26: As suggested, we removed the arrows for the sites with no trend in Fig. 2b for better clarity. Additionally, we increased the thickness of the lines in the legend of Fig. 2c to facilitate the distinction between colors.
- Fig. 3: Should the resolution be in meters or in square meters?
- R1.27: We thank Referee #1 for the observation. The spatial resolution is expressed in meters throughout the text (e.g., L191). It represents the length of the cell side in the gridded datasets, rather than the area. For consistency, we maintained this unit (meters) in the figure as well. To address the reviewer's concern, we have now clarified this in the caption of Fig. 3 (L341-342).

**REFEREE #2**

The authors investigated the spatiotemporal controls of soil moisture in boreal forest landscape in Sweden. They monitored surface soil moisture at 82 locations during a few months period. These soil moisture measurements were analyzed together with a vast array of environmental and hydrometeorological factors. Different spatial and temporal scales were considered. The study contributes to advancing models that represent spatiotemporal soil moisture variability.

Overall, the scope of the study is important and suitable for HESS. The methodology is clear and rather straightforward, and the number of considered predictors is impressive. The paper is generally well conceptualized and written. However, I have some suggestions to further improve the paper, especially the introduction and discussion.

We sincerely thank Referee #2 for outlining the main points of the paper, for the overall positive feedback, and for offering thoughtful critiques and suggestions presented below, which have enhanced the manuscript. Comments from the reviewer are in black and author replies are in blue, marked as R2.1-R2.15 for easy reference. Changes are also highlighted in light blue in the manuscript and supplement.

**General comments:**

In my opinion, the most concrete contribution that the study can make is to identify and provide key spatial and temporal predictors for data-driven models used in soil moisture mapping. While this aim and contribution is relevant and briefly mentioned, it would benefit from being more explicitly emphasized and clarified throughout the manuscript, particularly in the abstract, introduction, and discussion. Strengthening these sections could better position the study within the broader context of soil moisture research and mapping.

It remains unclear whether the study contributes to understanding the processes driving soil moisture variability. Therefore, it is also doubtful whether the study offers a contribution to the development of physically based models. Such models already incorporate key spatial (e.g., vegetation, soil texture, topography) and temporal (e.g., meteorological forcing) factors, and the manuscript does not address how the findings could enhance these models. Perhaps the authors could emphasize the need for more accurate field and remote sensing data on the identified variables, which could indirectly benefit physically based model approaches that rely on such data.

Although this study may not directly advance our understanding of the processes driving soil moisture variability, the identification of key predictors, many of which are already incorporated in physically based models, presents an opportunity for the authors to discuss how these different methods (process-based and data-driven) could be integrated for soil moisture mapping (e.g. hybrid approaches).

The amount of predictors and scales is impressive, but there are still many limitations to this study (as to any study). It is not a major problem, but the limitations need more attention in the discussion. For instance, topographic and tree shading is neglected, the spatial scales go only up to 30 or 64 m, and temporal extend was only a few months, just to name a few.

**R2.1: We thank Referee #2 for these valuable suggestions.**

- a) We have now clearly emphasized the aim and concrete contribution of this paper in the abstract (L15-16 and L30-31), introduction (L53-56 and L118), and discussion (L540-545) to better position our study within soil moisture research and mapping.
- b) We acknowledge that our study does not focus on the mechanisms driving soil moisture but rather on identifying key spatial and temporal predictors for data-driven models. As noted in R1.2a, we will reduce the emphasis on processes (L46, L53-56, L73-77, and L118). We also provided concrete examples (e.g., enhanced soil maps) of how our findings could benefit both data-driven and physically based approaches (L538-539).
- c) This is an interesting and relevant point. However, given the breadth of topics already covered and the considerable length of the manuscript, we decided not to delve into this discussion in detail. We agree that exploring hybrid approaches is a promising direction and worth pursuing in future work.
- d) We thoroughly discussed the key study limitations in a new section (L524-535), including the omission of microclimate spatial predictors (R1.4, R1.11), potential measurement errors, temporal discrepancies in data collection, inability of our models to explain the full spatial variation in soil moisture, and the restriction of our analysis to only one incomplete snow-free season (R2.4). Additionally, we addressed the lack of topographic shading by incorporating it into the analysis (see R1.2b, R2.9b).

**Specific comments:**

L33-34: Please add reference to "influence soil nitrogen availability"

R2.2: The reference to Nogovitcyn et al. (2023), which is already cited in the main text, covers both "soil nitrogen availability" and "needle production" (L36-37).

L36-39: It would help to have example reference to each of these applications that require soil moisture state.

R2.3: Thanks for the observation. To improve clarity, we have now added an example reference to explicitly match its corresponding application (L39-41).

L77-80: This is a good point that topography can have a major role in the early summer. In this study, you basically skipped the period impacted by snowmelt. How do you think the results would differ if full snow-free season was included?

R2.4: As suggested, we have now addressed this point when discussing the key study limitations (see R2.1d) (L32-33). Indeed, topography is likely to play a more significant role after snowmelt in our study area as well.

L83-84: I don't quite understand this sentence. You already described how the impact of topography changes between wet and dry seasons. Perhaps you mean that this may change within the season as well. This could be reformulated. Also please add references to the recent research you're referring to.

- R2.5: As both Referee #2 and Referee #1 (see R1.9) highlight, this sentence is unclear. To improve clarity, we rephrased this sentence and improved the connection with the following one (L89-92). The references for the recent research we are referring to were already included in the original version, so we kept them (L91-92).
- L135: How were the soil moisture locations decided? They mostly fall in the intensive monitoring area. How much of them are on peat soils? How much in mineral? Why this subset of the measurements?
- R2.6: We appreciate Referee #2's comment. As noted in R1.12, we did not use a formal sampling strategy (e.g., random or stratified sampling) but followed key principles now outlined in L159-167. Our primary goal was to mirror the distribution of soil moisture conditions representative of Swedish forests (please see R1.12 for a more detailed explanation and a graphic visualization). To achieve this, we placed about half of the loggers in the highly heterogeneous central part of the Krycklan catchment. Some loggers were positioned near permanent stations for comparison between different measurement systems, while the rest were distributed to balance spatial coverage and accessibility. We revised the manuscript to clarify the selection process (L159-167). Regarding the proportion of loggers in peat versus mineral soils, we added a new supplementary figure (S1) showing land use/land cover (Fig. S1d) and soil (Fig. S1a) information in relation to logger locations (see R1.3b and R2.9).
- L137: I was surprised not to see any fully saturated measurements on peat soils (peat soil porosity is around 0.90). Were there no peatlands (or peatlands with measurements) where the water table was near the surface? Hence, I don't think these measurements "capture the full spectrum of soil moisture levels across the Swedish landscape."
- R2.7: Thanks for this observation. Our loggers were placed across a wide range of soil moisture conditions, including very wet peat soils with water tables near the surface. However, our measurements of volumetric water content (VWC) range from 0 to 0.60 rather than 0 to 1 because VWC accounts for the total volume of soil, including solid particles, water, and air. Even in fully saturated peat, VWC does not reach 100% because soil particles occupy a portion of the total volume. The observed maximum values (~60%) are consistent with expectations for saturated peat soils, considering their high porosity (~90%) and the inherent structure of organic material. To clarify this point, we added a note in parenthesis in section 3.1 (L270).
- L167.: While citing for details is understandable, it would be good to elaborate a bit more on how the topographic indices were defined. For example, it makes a difference which method is used (e.g. D8 vs. Dinf). Please describe also the site quality index.
- R2.8: We thank Referee #2 for pointing this out. As the definition of variables not only the topographic indices but also all other predictors is a major shared concern with Referee #1 (see R1.2b and R1.3a), we provided a detailed description of each variable in the supplement, including the methods used to calculate the topographic indices. This ensures transparency and accuracy, allowing readers to interpret the results correctly. We revised the text accordingly (L199-200, L209, L222, and L228).
- Sect. 2.2.2: Please consider adding spatial predictor figure in supplement. It would give the reader a better idea how the landscape looks like. It is an impressive number of predictors, but one obvious one is missing: topographic shading.

**R2.9: Thanks for the suggestions.**

- a) We agree that adding maps illustrating some of the predictors would provide a better spatial context for the study landscape. As also suggested by Referee #1, we have now included a 4-panel map (Fig. S1) with information regarding soil (quaternary deposits, Fig. S1a), topography (elevation, Fig. S1b), vegetation (normalized difference vegetation index, Fig. S1c), and land use/land cover (Fig. S1d) in the supplementary material (see R1.3b).
- b) Both Referee #2 and Referee #1 (R1.2b) identified topographical variation in solar radiation, including the effect of topographic shading, as an important and easy-to-calculate variable that was missing in the previous version of the manuscript. To address this, we repeated the spatial OPLS analysis, this time incorporating both diffuse and direct solar radiation (calculated from the original DEM), accounting for the topographic shading effect. However, we excluded slope and aspect as their information is already included within the solar radiation variables (see detailed description in the Supplement). Since we rerun this analysis, we also took this opportunity to add two previously omitted variables: (1) the clear-cut class from the Land map, to match the analogous variable determined in the field, and (2) the glacifluvial sediment class from the SGU Quaternary deposit map, due to the inclusion of an additional layer of information (see detailed description in the Supplement). As a result, we revised Table 1, Figures 3, 5, S2, and S4, along with the corresponding text in the manuscript (e.g., L318-319).

Sect. 3.2. repeats many of the things already described in the methods (Sect. 2.3., which is great by they way). Perhaps the chapter in 3.2. can be integrated to 2.3.?

R2.10: Thanks for this comment. We acknowledge that the repetition concerns the first two sentences (L293-296) of section 3.2. However, we believe that it is important to repeat these details here to help readers interpreting figures 3 and 4. The two sections are substantially different. In Section 2.3, we provide the statistical background of the OPLS analysis, describe the different models created, and outline the metrics used for evaluation. In Section 3.2, however, our focus is on the visualization aspect, helping readers interpret the figures by explaining plot elements such as shapes, colors, and loading positions. This prevents repetitions when describing the results in the following subsections and is particularly useful for readers unfamiliar with the method. We believe that placing this explanation at the beginning of the results section, where the OPLS plots are presented, enhances readability.

L349: What is a vegetation period? Analysis starts from July something, growing season starts earlier.

R2.11: This is a good point. Indeed, the growing season starts earlier in Krycklan, making the term not entirely appropriate in this case. As mentioned in R1.5, we have now used broader terms like "study period" (L388) or "summer" (L17). When more detail was needed, we have specified "three snow-free months in 2022" (L110 and L434).

L370: "forest boreal landscape" -> "boreal forest landscape"

R2.12: Thanks for catching this mistake. We have now corrected it (L431).

L394: Kemppinen et al. study site (tundra) is quite different to Krycklan.

R2.13: We thank Referee #2 for the comment, which echoes a remark from Referee #1 (R1.19). As noted earlier, we have now deleted this comparison, and we added this reference at the beginning of the paragraph as an example of modeling soil moisture through LiDAR-derived topographic indices (L440).

L420: Consider replacing "performed poorly" with "correlated poorly with soil moisture" or something similar.

R2.14: As suggested, we replaced "performed poorly" with "showed weak correlation with soil moisture" (L480).

Code and data availability: Better practice would be to share the data in an openly available repository. Or has the data already been shared in the cited literature?

R2.15: We appreciate this suggestion. The data for most variables used in the analysis are already publicly available in the cited literature. However, the soil moisture time series from the TOMST loggers, their geographic locations within the Krycklan catchment, the field survey data listed in Table 1, and the topographic solar radiation data (added after this review) had not been published before. In response to the reviewer's recommendation, we have now made all of this data openly available in an online repository [https://doi.org/10.17632/s8zg5ymkh6.1]. We updated the corresponding references in Table 1 and revised the data availability statement accordingly (L558-560).

---

## Author Response (AR2)

Dear Dr. Toth,

Thank you for your note and for handling the review process, and thanks also to the reviewers for their helpful final comments.

We have implemented both remaining suggestions:

- The text has been revised as suggested by Referee #1 for improved clarity (L102 and L333).
- A sentence has been added in Section 4.4 (L527-529) explicitly acknowledging the calibration challenges in organic-rich peat soils, as recommended by Referee #2. The previous sentence (L525-527) has been adjusted to improve the flow and connection between the two sentences.

Best regards, Francesco Zignol on behalf of all co-authors